# Flower/FLWR-1 regulates neuronal activity via the plasma membrane Ca²⁺ ATPase to promote recycling of synaptic vesicles

**Marius Seidenthal[1,2], Jasmina Redzovic[1,2,3†], Jana F Liewald[1,2†], Dennis Rentsch[1,2], Stepan Shapiguzov[1,2], Noah Schuh[1,2], Nils Rosenkranz[1,2], Stefan Eimer[3], Alexander Gottschalk[1,2]\***

[1]Buchmann Institute for Molecular Life Sciences, Goethe-University, Frankfurt, Germany; [2]Institute for Biophysical Chemistry, Department of Biochemistry, Chemistry, and Pharmacy, Goethe-University, Frankfurt, Germany; [3]Institute of Cell Biology and Neuroscience, Goethe-University, Frankfurt, Germany

**\*For correspondence:**
a.gottschalk@em.uni-frankfurt.de

†These authors contributed equally to this work

**Competing interest:** The authors declare that no competing interests exist.

## eLife Assessment

This **important** study uses *C. elegans* to provide new insights into the role of the conserved protein FLWR-1/Flower in synaptic transmission. Employing a variety of techniques, including calcium imaging, ultrastructural analysis, and electrophysiology, the paper provides **convincing** evidence that challenges some previous thinking about FLWR-1 function. This work will be of particular interest to neuroscientists studying synaptic physiology and plasticity.

**Abstract** The Flower protein was suggested to couple the fusion of synaptic vesicles (SVs) to their recycling in different model organisms. It is supposed to trigger activity-dependent bulk endocytosis by conducting $Ca^{2+}$ at endocytic sites. However, this mode of action is debated. Here, we investigated the role of the *Caenorhabditis elegans* homologue FLWR-1 in neurotransmission. Our results confirm that FLWR-1 facilitates the recycling of SVs at the neuromuscular junction (NMJ). Ultrastructural analysis of synaptic boutons after hyperstimulation revealed an accumulation of large endocytic structures in *flwr-1* mutants. These findings do not support a role of FLWR-1 in the formation of bulk endosomes but rather a function in their breakdown. Unexpectedly, the loss of FLWR-1 led to increased neuronal $Ca^{2+}$ levels in axon terminals during stimulation, particularly in GABAergic motor neurons, causing excitation-inhibition imbalance. We found that this increased NMJ transmission might be caused by deregulation of MCA-3, the nematode orthologue of the plasma membrane $Ca^{2+}$ ATPase (PMCA). *In vivo* molecular interactions indicated that FLWR-1 may be a positive regulator of the PMCA and might influence its recycling through modification of plasma membrane levels of phosphatidylinositol-4,5-bisphosphate $(PI(4,5)P_2)$.

## Introduction

Chemical synaptic transmission involves a cycle of biogenesis of synaptic vesicles (SVs), their fusion with the plasma membrane (PM), as well as their recycling by endocytosis and *de novo* formation in the endosome (*Alabi and Tsien, 2012*; *Chanaday et al., 2019*; *Kononenko and Haucke, 2015*; *Rizzoli, 2014*; *Saheki and De Camilli, 2012*). Coupling of SV exocytosis and endocytosis must be tightly controlled to avoid depletion of the reserve pool of SVs and to enable sustained neurotransmission

(*Haucke et al., 2011*; *Lou, 2018*). Different hypotheses have been formulated as to how this is achieved within neurons. One hypothesis suggests that an SV-integral transmembrane protein called Flower could form ion channels that get inserted into the PM during SV fusion (*Yao et al., 2009*). Subsequently, Flower may facilitate endocytosis by conducting $Ca^{2+}$ into the cytoplasm, thus contributing to defining endocytic sites (*Yao et al., 2017*). In *Drosophila melanogaster*, Flower was proposed to increase phosphatidylinositol-4,5-bisphosphate $(PI(4,5)P_2)$ levels through $Ca^{2+}$ microdomains, which was suggested to drive activity-dependent bulk endocytosis (ADBE) and formation of new SVs after prolonged, intense neurotransmission (*Li et al., 2020*). The hypothesis that Flower may have $Ca^{2+}$ channel activity was further proposed based on sequence similarities between Flower and the $Ca^{2+}$ selectivity filter of voltage-gated $Ca^{2+}$ channels (VGCCs; *Yao et al., 2009*). Additional evidence from *Drosophila* suggests that Flower may regulate clathrin-mediated endocytosis in a $Ca^{2+}$-independent fashion (*Yao et al., 2017*). However, while a facilitatory role of Flower in endocytosis appears to be evolutionarily conserved and was observed in different organisms and tissues, including non-neuronal cells (*Chang et al., 2018*; *Rudd et al., 2023*; *Yao et al., 2017*), its $Ca^{2+}$ channel activity and its influence on SV recycling is debated (*Coelho and Moreno, 2020*; *Lou, 2018*). The kinetics of $Ca^{2+}$ rise mediated by Flower appear to be too slow, and the amount conducted is too low to have a major impact (*Xue et al., 2012*). Moreover, the function of Flower appears to depend on the cell type and the extent of synaptic activity (*Chang et al., 2018*; *Yao et al., 2017*). Apart from its role in endocytosis, Flower is involved in cell survival mechanisms during development (*Coelho and Moreno, 2020*; *Costa-Rodrigues et al., 2021*). Intriguingly, loss of the mammalian homologue of Flower can reduce tumor growth, suggesting an important function in tumor cell survival and indicating possible options for cancer treatment (*Madan et al., 2019*; *Petrova et al., 2012*). However, more research is needed to determine the exact signaling pathways by means of which Flower mediates cell survival (*Costa-Rodrigues et al., 2021*).

Early studies of the Flower protein proposed a two- or three-transmembrane helical organization in which the C-terminus is exposed to the extracellular space and may thus mediate intercellular communication (*Costa-Rodrigues et al., 2021*; *Rhiner et al., 2010*). However, more recent research has shown that both N- and C-termini of Flower are likely cytosolic, and that the longest mammalian isoform consists of four transmembrane helices which are connected by short loops (*Chang et al., 2018*; *Rudd et al., 2023*). The genome of the nematode *Caenorhabditis elegans* is predicted to contain only a single isoform of Flower (FLWR-1; wormbase.org). *C. elegans,* therefore, may serve as an excellent model organism to further investigate the evolutionarily conserved role of Flower in neurotransmission and endocytosis. Here, we studied the function of *C. elegans* FLWR-1. We find that FLWR-1 localizes to SVs and to the PM and is involved in neurotransmission. We further show that FLWR-1 has a facilitatory but not essential role in endocytosis, confirming previous research in *Drosophila* and mammalian cells. Loss of FLWR-1 surprisingly conveys increased $Ca^{2+}$ levels in optogenetically depolarized motor neurons. Yet, this is accompanied by reduced neurotransmitter release based on pharmacological assays and following optogenetic stimulation, thus suggesting a deregulation of $Ca^{2+}$ signaling in the presynapse. This is associated with defective SV recycling at the level of the endosome and thus reduced SV numbers upon sustained stimulation. The increased $Ca^{2+}$ level is more pronounced in γ-aminobutyric acid (GABA) releasing neurons and leads to an excitation-inhibition (E/I) imbalance at the neuromuscular junction (NMJ) through increased release of the neurotransmitter GABA. A function of FLWR-1 in endocytosis appears to also affect the PM $Ca^{2+}$ ATPase MCA-3, required for extrusion of $Ca^{2+}$ from the cytosol. This may explain the increased $Ca^{2+}$ levels during stimulation in *flwr-1* mutant neurons. Lastly, our findings suggest a possible direct molecular interaction between FLWR-1 and MCA-3.

## Results

### FLWR-1 is involved in neurotransmission

The amino acid sequence of *C. elegans* FLWR-1 is conserved with its *D. melanogaster* (Fwe-Ubi/ FweA) and *Homo sapiens* (hFwe4) homologues (*Figure 1A*). Fwe-Ubi/FweA was shown to facilitate recovery of neurons, i.e., refilling of SV pools, following intense synaptic activity (*Yao et al., 2009*; *Yao et al., 2017*). To investigate whether this involvement of Flower in neurotransmission is evolutionarily conserved, we studied a mutant lacking most of the *flwr-1* coding region (*ok3128*, *Figure 1B*; *C.*

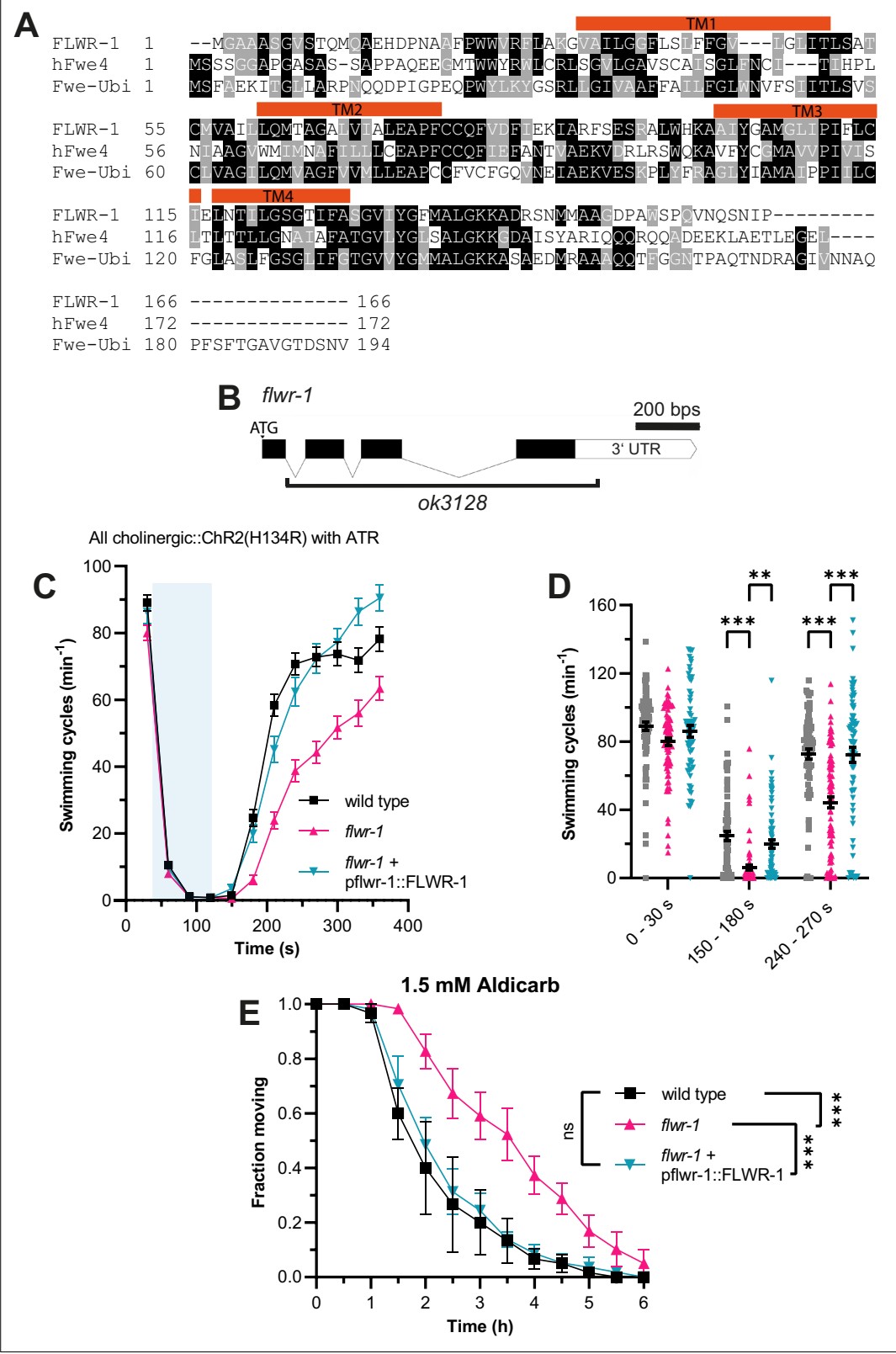

**Figure 1.** Loss of FLWR-1 induces defects in neurotransmission following intense stimulation. (**A**) Alignment of the amino acid sequences of FLWR-1 to hFwe4 (*H. sapiens*) and Fwe-Ubi/FweA (*D. melanogaster*). Shading depicts evolutionary conservation of amino acid residues (identity – black; homology – gray). Position of TM helices indicated in red refers to the FLWR-1 sequence. (**B**) Schematic representation of the *flwr-1*/F20D1.1 gene

*Figure 1 continued on next page*

*Figure 1 continued*

locus and the size of the *ok3128* deletion. Bars represent exons and connecting lines introns. (**C**) Mean (± SEM) swimming cycles of animals expressing ChR2(H134R) in cholinergic motor neurons (*unc-17* promoter). All animals were treated with all-*trans* retinal (ATR). A 90 s light pulse (470 nm, 1 mW/mm$^2$) was applied after 30 s as indicated by the blue shade. Number of animals accumulated from N=3 biological replicates: wild type = 80–88, *flwr-1* = 80–91, FLWR-1 rescue = 62–75. (**D**) Statistical analysis of swimming speed at different time points as depicted in (**C**). Mean (± SEM). Each dot represents a single animal. Mixed-effects model analysis with Tukey's correction. Only statistically significant differences are depicted. **$p<0.01$, ***$p<0.001$. (**E**) Mean (± SEM) fraction of moving animals after exposure to 1.5 mM aldicarb. N=4 biological replicates. Two-way ANOVA with Tukey's correction. ns, not significant, $p>0.05$, ***$p<0.001$.

The online version of this article includes the following source data and figure supplement(s) for figure 1:

**Source data 1.** Raw data for *Figure 1C-E*.

**Figure supplement 1.** Loss of FLWR-1 does not change body length or locomotion speed but reduces the number of living progeny.

**Figure supplement 1—source data 1.** Raw data for *Figure 1—figure supplement 1A-C*.

**Figure supplement 2.** FLWR-1 is predicted to be a tetraspan transmembrane protein and is transported by UNC-104 kinesin.

**Figure supplement 2—source data 1.** Raw data for *Figure 1—figure supplement 2D and E*.

**Figure supplement 3.** FLWR-1 expression in body wall muscle (BWM) cells partially rescues aldicarb resistance of *flwr-1* mutants, but aldicarb resistance is not affected through ACh receptors (AChRs).

**Figure supplement 3—source data 1.** Raw data for *Figure 1—figure supplement 3A-D*.

*elegans Deletion Mutant Consortium, 2012*). Animals lacking FLWR-1 did not display severe phenotypes. Basal locomotion in liquid (*Figure 1C and D*, seconds 0–30) and body length of young adults (*Figure 1—figure supplement 1A*) were not significantly different from the respective values of wild type animals. However, fertility appeared slightly reduced (*Figure 1—figure supplement 1B*). Basal crawling speed, on average, was not different between wild type, *flwr-1* mutants, and rescued animals (genomic sequence and 2 kB promoter; *Figure 1—figure supplement 1C*).

These findings indicate that FLWR-1 is unlikely to have an essential function in neurotransmission, at least not during basal *in vivo* activity, but rather a regulatory or facilitatory one. Previously, we showed that mutations which only weakly affect basal locomotion can severely affect recovery of swimming speed after strong optogenetic stimulation of cholinergic neurons (*Yu et al., 2018*). We therefore subjected *flwr-1(ok3128)* mutants expressing channelrhodopsin-2 (ChR2; variant H134R), which were treated with the chromophore all-*trans* retinal (ATR), to blue light during swimming (*Liewald et al., 2008*; *Nagel et al., 2005*). Photostimulation resulted in a stop of all swimming during the light pulse, followed by a slow recovery in the dark. Indeed, the loss of FLWR-1 led to a significantly slowed recovery of swimming locomotion (*Figure 1C and D*). This could be fully rescued by transgenic expression of genomic *flwr-1* including a 2 kb sequence upstream of the putative start codon, hereafter called *flwr-1p* (as the promoter of *flwr-1*). To further evaluate the involvement of FLWR-1 in neurotransmission, we exposed *flwr-1* mutants to aldicarb. This acetylcholine esterase inhibitor induces paralysis due to the accumulation of ACh in the synaptic cleft (*Blazie and Jin, 2018*; *Mahoney et al., 2006*). Resistance to aldicarb indicates either reduced release or detection of ACh or, alternatively, increased inhibitory signaling (*Jánosi et al., 2024*; *Vashlishan et al., 2008*). Loss of FLWR-1 led to a significant delay in paralysis, indicating an involvement in transmission at the NMJ (*Figure 1E*). Full rescue of the *flwr-1* mutant phenotypes suggests that expression from the 2 kb fragment of the endogenous promoter fully recapitulates the native expression in the context of NMJ function.

## FLWR-1 is expressed in excitable cells and localizes to SVs

Invertebrate and vertebrate homologues of FLWR-1 are predicted to be membrane proteins containing four transmembrane helices with both C- and N-termini located in the cytosol (*Chang et al., 2018*; *Rudd et al., 2023*; *Yao et al., 2009*). *In silico* predictions using the FLWR-1 sequence of 166 amino acids suggest evolutionary conservation of the tetraspan structure (*Figure 1—figure supplement 2A*; *Hallgren et al., 2022*). Accordingly, AlphaFold3 (AF3) predicts a protein structure of FLWR-1 that implies a four-helical configuration, with helix lengths that could span biological membranes,

and three additional, shorter α-helices (*Figure 1—figure supplement 2B*; *Abramson et al., 2024*). We sought to determine the cellular as well as subcellular localizations of FLWR-1 by tagging its N-terminus with GFP and expressing the construct using the endogenous promoter (*Figure 2A–C*). Green fluorescence could be observed in developing embryos in the uterus, as well as in various tissues, including neurons, body wall, and pharyngeal muscles (*Figure 2A*). FLWR-1 localized to neurites and cell bodies of nerve ring neurons, representing the central nervous system of the nematode (*Figure 2B*; *Ward et al., 1975*). These results are in agreement with single-cell RNAseq data obtained from *C. elegans,* which show near-ubiquitous expression of *flwr-1/F20D1.1* (*Figure 1—figure supplement 2C*, *Supplementary file 3*; *Taylor et al., 2021*). The GFP::FLWR-1 fusion protein was functional as demonstrated by the rescue of the swimming phenotype (recovery from cholinergic neuron photostimulation) observed in the *flwr-1* mutant background (*Figure 1—figure supplement 2D and E*).

Within muscle cells, FLWR-1 was primarily targeted to the PM (*Figure 2C*). As its *D. melanogaster* homologue was localized to SVs (*Yao et al., 2009*), we wondered whether FLWR-1 would colocalize with known SV markers such as SNB-1 synaptobrevin-1 (*Calahorro and Izquierdo, 2018*; *Nonet, 1999*). Indeed, GFP::FLWR-1 fluorescence largely overlapped with co-expressed mCherry::SNB-1 in the dorsal nerve cord (DNC; *Figure 2D and E*). FLWR-1 further seemed to be enriched in fluorescent puncta in DNC and sublateral nerve cords, indicating synaptic localization (*Figure 2D–F*); however, it was not restricted to synaptic regions only, meaning it is likely present also in the PM. Furthermore, we observed moving particles, probably SV precursors, which contained SNB-1 and FLWR-1, traveling along commissures between ventral nerve cord (VNC) and DNC (*Figure 2F and G* and *Video 1*). Anterograde transport of these precursors toward synapses depends on kinesin-3/UNC-104 (*Hall and Hedgecock, 1991*; *Klopfenstein and Vale, 2004*). We used a reduction-of-function allele (*e1265*) affecting the interaction of UNC-104 with its cargo to investigate whether FLWR-1 is actively transported toward synapses (*Cong et al., 2021*). Indeed, animals lacking functional UNC-104 showed a reduced amount of axonal GFP fluorescence in the nerve ring and DNC while cell bodies were clearly visible (*Figure 1—figure supplement 2F*). The ratio of DNC to VNC fluorescence was significantly decreased in *unc-104* mutants, suggesting defective anterograde transport (*Figure 2H*). Distribution of SNB-1 was similarly affected (*Cuentas-Condori et al., 2023*; *Gally and Bessereau, 2003*). Together, these results argue that FLWR-1 is expressed in neurons (as well as in muscles and other cell types) and is actively transported toward synapses.

## GABAergic signaling is increased in *flwr-1* knockout mutants

Since FLWR-1 is also expressed in body wall muscles (BWMs), we wondered whether the observed aldicarb resistance originates from reduced ACh reception by ACh receptors (AChRs) (*Mahoney et al., 2006*). To test this, we exposed animals lacking FLWR-1 to levamisole, an AChR agonist which induces paralysis by hyperexcitation and body contraction, similar to aldicarb (*Gottschalk et al., 2005*; *Sattelle et al., 2002*). We found no significant differences in the rate of paralysis between wild type and mutant animals (*Figure 1—figure supplement 3A*). However, transgenic expression of FLWR-1 in BWMs partially rescued the aldicarb resistance (*Figure 1—figure supplement 3B*). Therefore, the loss of FLWR-1 might decrease the ACh response of muscle cells, which may contribute to the aldicarb resistance of *flwr-1* mutants. However, as muscles can also affect motor neurons by inhibitory retrograde signaling (*Hu et al., 2012*; *Simon et al., 2008*; *Tong et al., 2017*), we wanted to assess if the loss of FLWR-1 in muscle may also have effects on motor neurons. Thus, we rescued FLWR-1 in muscle and either cholinergic or GABergic neurons (*Figure 1—figure supplement 3B*). The combined rescues did not show any additive effects to the pure neuronal rescues; thus, FLWR-1 effects on muscle cell responses to cholinergic agonists might be cell-autonomous. Yet, as cholinergic rescue alone did not overcompensate *flwr-1* mutants' resistance in aldicarb assays (*Figure 3D*), this interpretation is complicated. Since muscles are activated by ACh via two different AChRs, the levamisole receptor and the nicotine-sensitive receptor (N-AChR), a homopentamer of ACR-16 subunits (*Almedom et al., 2009*; *Richmond and Jorgensen, 1999*), we addressed the possibility that FLWR-1 may regulate the expression or function of N-AChRs in muscle, to affect phenotypes of aldicarb resistance. We performed the aldicarb assay in the absence of ACR-16 (*Figure 1—figure supplement 3C*), which showed that the two mutations, *flwr-1(ok3128)* and *acr-16(ok789)*, had additive effects. Thus, FLWR-1 does not affect aldicarb resistance through downregulation of nAChRs, as otherwise the double mutant would not be more resistant than the *flwr-1* single mutant.

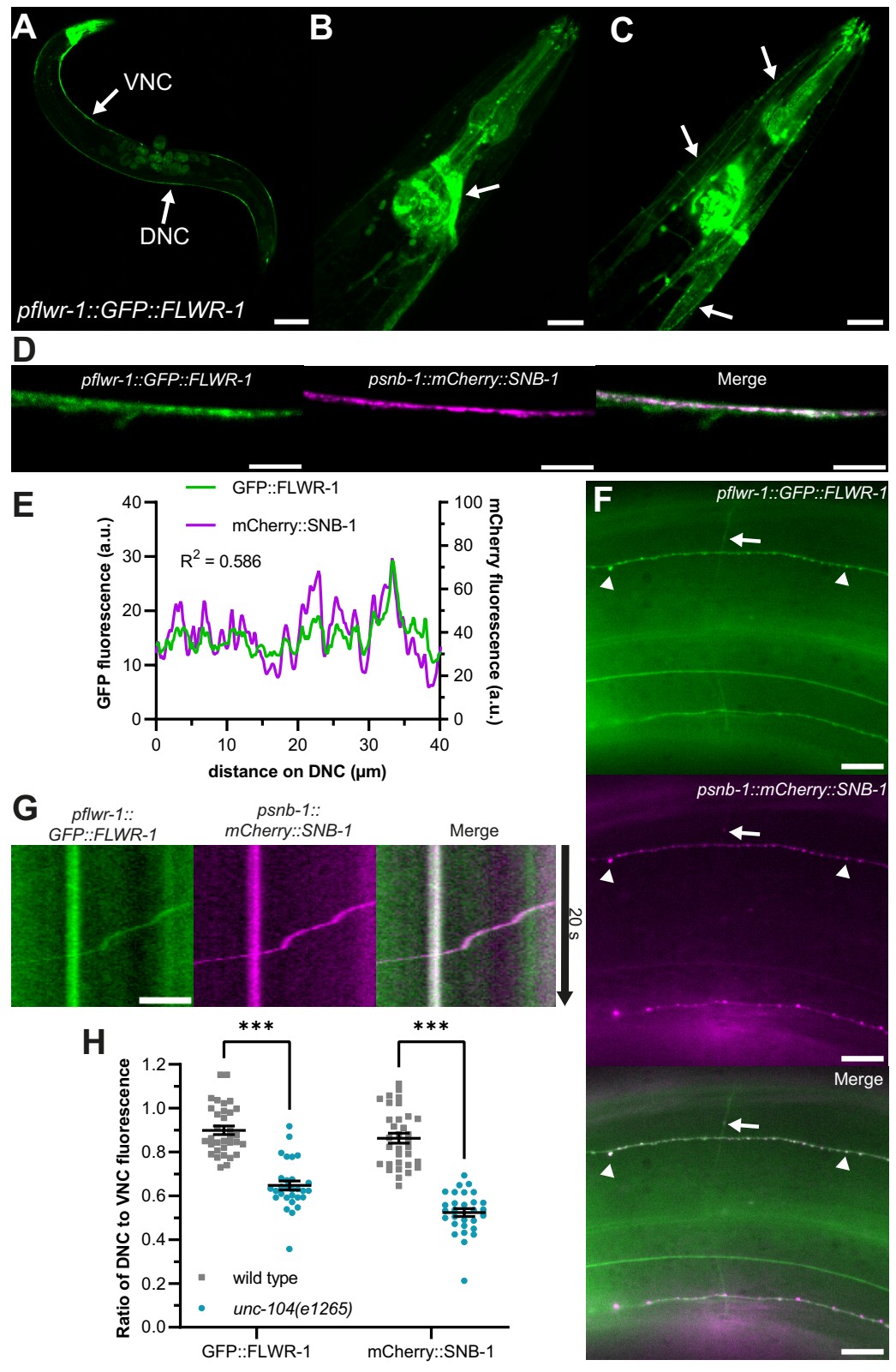

**Figure 2.** FLWR-1 is expressed in neurons and localizes to synaptic vesicles and the plasma membrane.
(**A–D**) Confocal micrographs (maximum projection of z-stacks or single plane) of animals co-expressing *flwr-1p::GFP::FLWR-1* and *psnb-1::mCherry::SNB-1*. (**A**) Overview of GFP::FLWR-1 expression. Arrows indicate dorsal and ventral nerve cords (DNC and VNC, respectively). Scale bar, 100 µm. (**B**) GFP::FLWR-1 in head neurons and

*Figure 2 continued on next page*

*Figure 2 continued*

pharynx. Scale bar, 20 µm. (**C**) Animal depicted in (**B**), single plane showing neck muscle cells. Arrows indicate GFP::FLWR-1 localization to the plasma membrane. Scale bar, 20 µm. (**D**) GFP and mCherry fluorescence in the DNC. Scale bar, 10 µm. (**E**) Line scan analysis of colocalization of GFP::FLWR-1 and mCherry::SNB-1 along the DNC as represented in (**D**). $R^2$ as determined by Pearson correlation. a.u.=arbitrary units of fluorescence intensity. (**F**) Micrograph depicting GFP::FLWR-1 and mCherry::SNB-1 fluorescence in sublateral nerve cords and commissures. Arrowheads indicate synaptic puncta. Arrow points toward synaptic vesicle (SV) precursor traveling along commissure as shown in *Video 1*. Scale bar, 10 µm. (**G**) Kymograph representing the SV precursor indicated in (**F**) traveling along commissures. Scale bar, 2 µm. (**H**) Comparison of the ratio of DNC to VNC fluorescence of GFP::FLWR-1 and mCherry::SNB-1 in wild type and *unc-104(e1265)* mutant background. Mean (± SEM). Each dot represents a single animal. Two-way ANOVA with Šídák's correction. \*\*\*p<0.001. Number of animals imaged in N=3 biological replicates: wild type = 33, *unc-104* = 29.

The online version of this article includes the following source data for figure 2:

**Source data 1.** Raw data for *Figure 2*.

In addition to muscle expression, we also observed enrichment of FLWR-1 in fluorescent puncta, indicating presynaptic localization (*Figure 3A and B*). Moreover, FLWR-1 partially colocalized with the dense projection marker ELKS-1 in cholinergic as well as in GABAergic neurons (*Figure 3A–C*; *Figure 3—figure supplement 1A–C*; *Dai et al., 2006*; *Kittelmann et al., 2013a*). Unlike ELKS-1, FLWR-1 could also be found in intersynaptic regions, though to a lesser extent than in synapses. This indicates that FLWR-1 might be involved in neurotransmission at both cholinergic and GABAergic NMJs, yet is not exclusively localized to active zones. The expression of FLWR-1 relative to ELKS-1 in either neuron type was not obviously biased to GABAergic or cholinergic neurons (*Figure 3—figure supplement 1A–C*). Thus, to determine whether the site of action of FLWR-1 in NMJ signaling is also evenly located to cholinergic and GABAergic neurons, we rescued FLWR-1 in each cell type of *flwr-1* mutants, using the promoters of the vesicular transporters of ACh (UNC-17) or GABA (UNC-47), respectively. Surprisingly, the expression of FLWR-1 in GABAergic, but not in cholinergic neurons, rescued aldicarb resistance (*Figure 3D*). This was unexpected, since mutations affecting SV recycling commonly lead to a slowed replenishment of SVs and thus reduced ACh release (*Salcini et al., 2001*; *Schuske et al., 2003*; *Yu et al., 2018*). However, our results indicate that loss of FLWR-1 leads to increased release of GABA, which counteracts the aldicarb-induced paralysis (*Câmara et al., 2019*). Additional expression of FLWR-1 in BWMs did not change these results, suggesting that its role in neurons is more crucial in affecting aldicarb sensitivity (*Figure 1—figure supplement 3B*). To confirm this, we crossed *flwr-1* mutants to animals lacking the vesicular GABA transporter UNC-47. This abolishes GABA release and induces hypersensitivity to aldicarb (*Vashlishan et al., 2008*). We observed that the additional mutation of *unc-47* led to a complete loss of aldicarb resistance in *flwr-1* mutants (*Figure 3E*). No difference between *unc-47* and *flwr-1; unc-47* double mutants could be observed. However, as we used a high concentration of aldicarb, possible additional effects of the double mutant may have been masked. Thus, we also tested these animals at a lower concentration of aldicarb. *unc-47; flwr-1* double mutants were hypersensitive to aldicarb compared to *unc-47* single mutants, suggesting that ACh release is also increased by the loss of FLWR-1 (*Figure 1—figure supplement 3D*). In the absence of the compensatory increase in GABAergic signaling, this conveys aldicarb hypersensitivity. Jointly, these findings support the notion that it is primarily increased GABA signaling which is the main cause of aldicarb resistance in *flwr-1* mutants, yet neurotransmission in motor neurons may be generally upregulated.

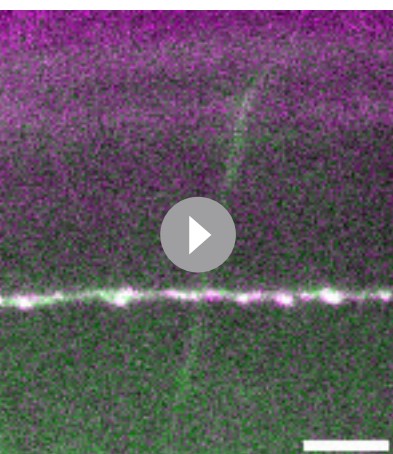

**Video 1.** Time series of a particle containing GFP::FLWR-1 and mCherry::SNB-1 traveling along commissures. Scale bar, 2 µm.

https://elifesciences.org/articles/103870/figures#video1

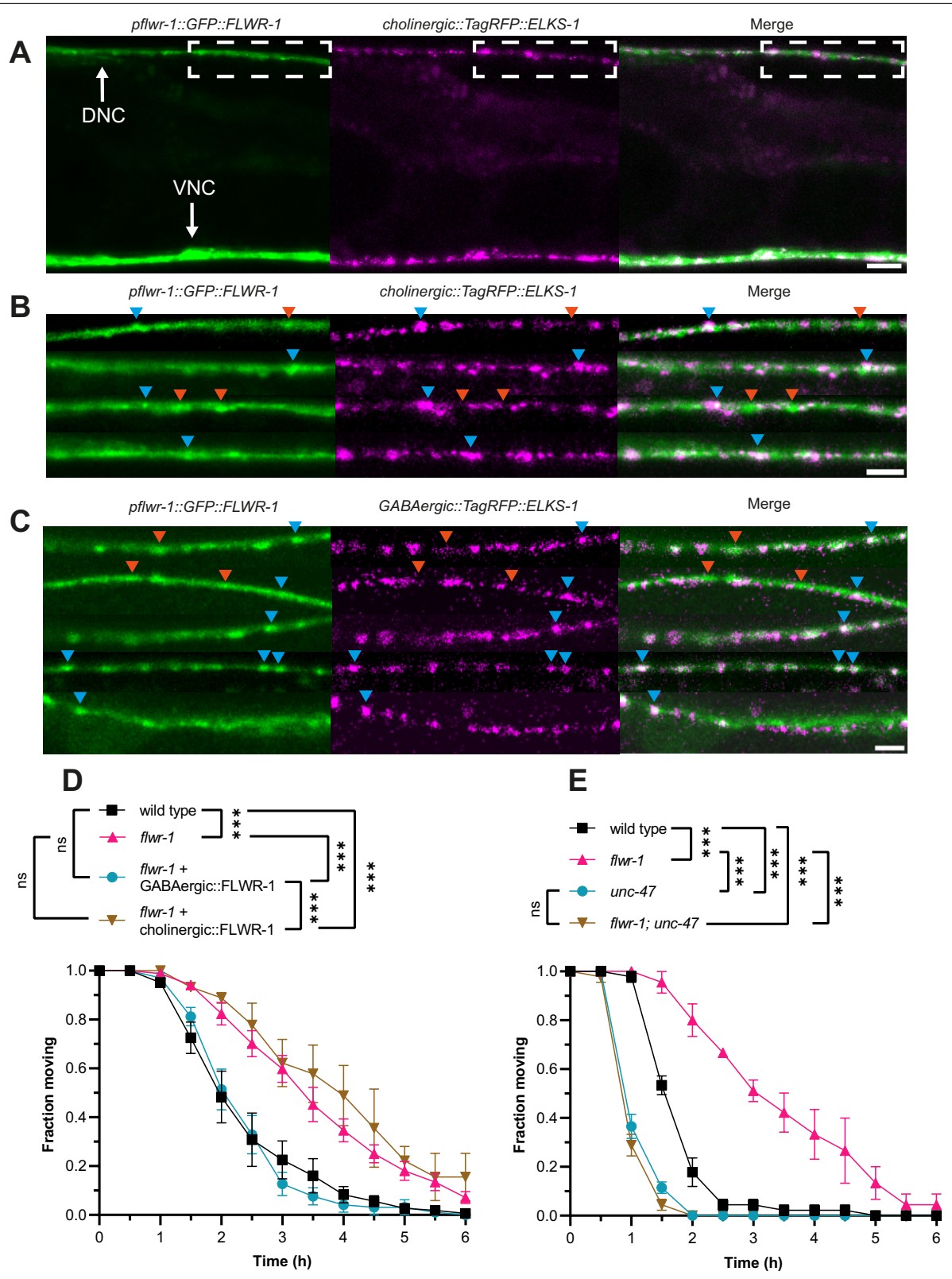

**Figure 3.** GABAergic signaling is increased in *flwr-1* knockout mutants. (**A**) Representative confocal micrographs of an animal co-expressing *flwr-1p::GFP::FLWR-1* and *TagRFP::ELKS-1* in cholinergic motor neurons (*unc-17(short)* promoter). The region in the dorsal nerve cord (DNC) used to acquire images shown in (**B**) and (**C**) is indicated; this position is posterior to the vulva, anterior is left. Scale bar, 10 μm. (**B, C**) DNCs in animals co-expressing *flwr-1p::GFP::FLWR-1* and *TagRFP::ELKS-1* in cholinergic motor neurons (*unc-17(short)* promoter) or in GABAergic neurons (*unc-47* promoter;

*Figure 3 continued on next page*

*Figure 3 continued*

**C**), respectively. Scale bar, 5 µm. Examples of fluorescent puncta which contain either both FLWR-1 and ELKS-1 (blue arrowheads), or FLWR-1 only (red arrowheads) are indicated. The same puncta are indicated in the respective analysis of signal density along the DNC in *Figure 3—figure supplement 1A and B*. (**D**) Mean (± SEM) fraction of moving animals after exposure to 1.5 mM aldicarb with cholinergic (*unc-17p*) and GABAergic (*unc-47p*) expression of FLWR-1 in *flwr-1(ok3128)* mutant background. N=3–8 biological replicates. (**E**) Mean (± SEM) fraction of moving animals after exposure to 1.5 mM aldicarb with *unc-47(e307)* and *unc-47(e307); flwr-1(ok3128)* double mutants. N=3 biological replicates. Two-way ANOVA with Tukey's correction in D, E. ns, not significant, ***p<0.001.

The online version of this article includes the following source data and figure supplement(s) for figure 3:

**Source data 1.** Raw data for *Figure 3D and E*.

**Figure supplement 1.** FLWR-1 localizes to cholinergic and GABAergic active zones.

**Figure supplement 1—source data 1.** Raw data for *Figure 3—figure supplement 1A-C*.

## Depolarization-induced neuronal Ca²⁺ is increased in *flwr-1* mutants

Since the release of GABA appeared to be increased in *flwr-1* mutants, we wondered whether the response of GABAergic neurons during activation was increased. To test this, the fluorescent Ca²⁺ indicator GCaMP was expressed in GABAergic neurons, allowing us to estimate relative Ca²⁺ levels at presynaptic sites (*Lu et al., 2022*; *Nakai et al., 2001*). To allow optogenetic depolarization of neurons independent of GCaMP excitation light, we co-expressed the red-shifted channelrhodopsin variant ChrimsonSA (*Oda et al., 2018*; *Seidenthal et al., 2022*). As expected, optogenetic stimulation caused an increase in GCaMP fluorescence at NMJs in the DNC (*Figure 4A*). Comparing wild type and *flwr-1* mutants, we found that the gain in fluorescence intensity was significantly higher in animals lacking FLWR-1 (*Figure 4A and B*), supporting our finding of increased neurotransmission in GABAergic neurons. To verify this at the behavioral level, we used a strain expressing ChR2(H134R) in GABAergic neurons, as it can be used to assess GABA release through measurement of body length (*Liewald et al., 2008*). Since GABA receptors hyperpolarize muscle cells, optogenetic stimulation of GABAergic motor neurons causes relaxation and thus increased body length (*Schultheis et al., 2011*; *Seidenthal et al., 2022*). To augment the effect of ChR2 stimulation on body length and to observe the effect of GABA release independent of cholinergic neurotransmission, the assay was performed in a mutant background lacking the levamisole receptor (*unc-29(e1072)* mutant, affecting an essential subunit; *Fleming et al., 1997*; *Richmond and Jorgensen, 1999*). As expected, stimulation with blue light led to increased body length (*Figure 4C*). In accordance with Ca²⁺ imaging results, *flwr-1; unc-29* double mutants showed a significantly stronger elongation during stimulation than *unc-29* mutants (*Figure 4C and D*). These results indicate that Ca²⁺ influx into the synaptic cytosol is increased, likely causing more GABA to be released in animals lacking FLWR-1. The previously observed aldicarb resistance indicates that this change in GABAergic transmission outweighs possible changes in cholinergic transmission (*Figure 3*).

To investigate whether neuronal responses to depolarization may be generally upregulated in *flwr-1* mutants, we assessed whether evoked Ca²⁺ level increase is affected in cholinergic motor neurons as well (*Figure 4E and F*). Again, we observed increased GCaMP fluorescence levels during optogenetic stimulation. This effect could be rescued in cholinergic neurons. However, since postsynaptic muscle exerts inhibitory retrograde signaling to presynaptic cholinergic neurons (*Hu et al., 2012*; *Simon et al., 2008*; *Tong et al., 2017*), and because FLWR-1 is also expressed in muscle, we tested if the phenotype of the loss of FLWR-1 in cholinergic neurons would be affected by rescuing FLWR-1 in muscle. This was not the case (*Figure 4E and F*). Together, these results indicate that loss of FLWR-1 conveys an upregulation of neuronal Ca²⁺ level rise during continuous stimulation in both classes of motor neurons (*Figure 4G*). However, the overall E/I balance appears to be shifted toward stronger inhibition of BWMs.

## Endocytosis is slowed in *flwr-1* mutants in non-neuronal cells and neurons

Homologues of FLWR-1 were implicated in endocytosis in neurons as well as in non-neuronal cells (*Chang et al., 2018*; *Rudd et al., 2023*; *Yao et al., 2009*). We thus wondered whether this function is evolutionarily conserved in nematodes. One possibility to assess this in *C. elegans* is to observe endocytosis in coelomocytes (CCs; *Fares and Greenwald, 2001*). These scavenger cells continuously

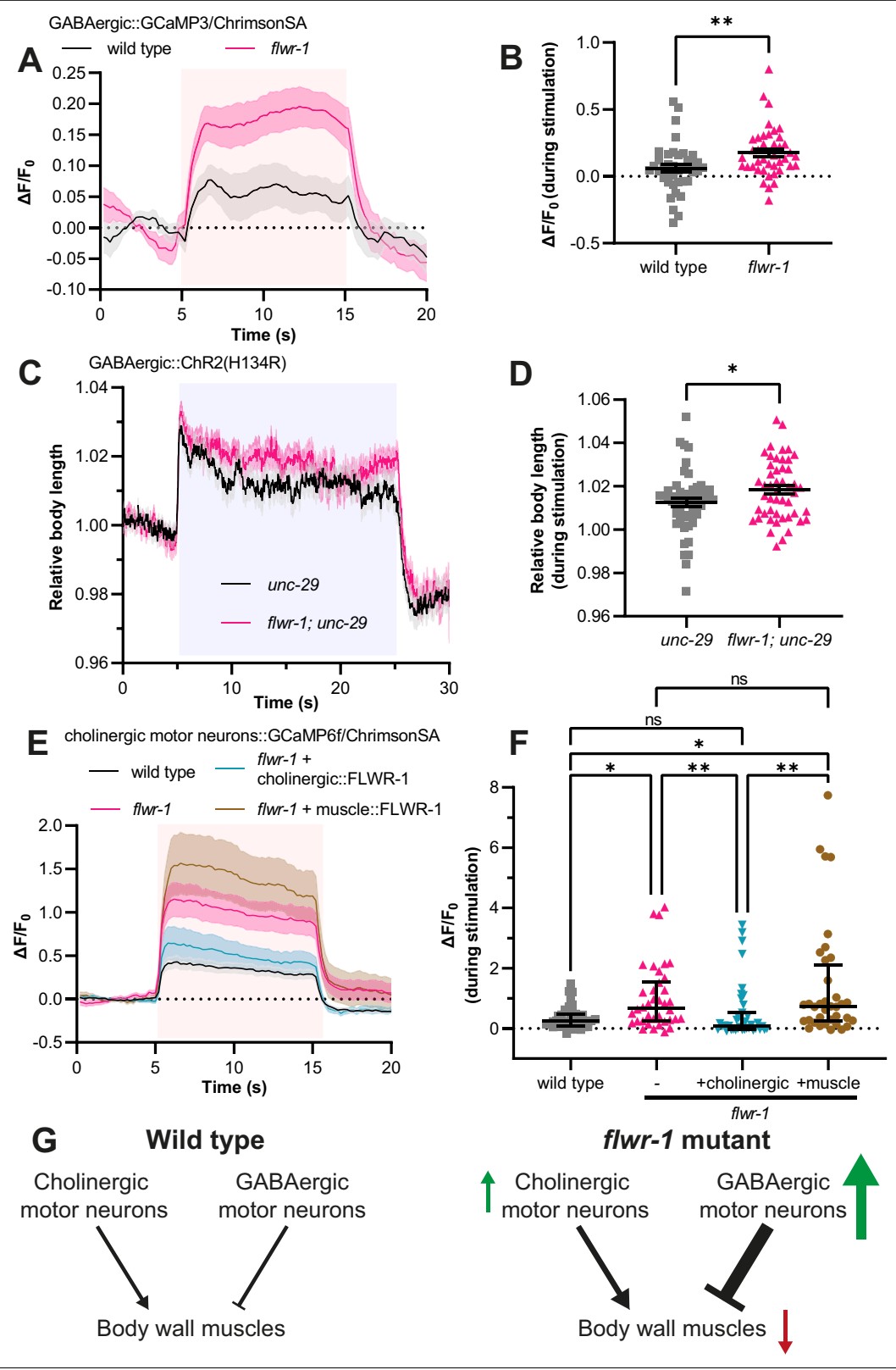

**Figure 4.** Loss of FLWR-1 leads to increased Ca²⁺ levels during stimulation. (**A**) Mean (± SEM) normalized fluorescence in synaptic puncta of the dorsal nerve cord (DNC) of animals expressing GCaMP3 and ChrimsonSA in GABAergic neurons (*unc-25* promoter). All animals were supplemented with all-*trans* retinal (ATR). A 10 s light pulse (590 nm, 40 µW/mm²) was applied after 5 s as indicated by the red shade. (**B**) Mean (± SEM) normalized

*Figure 4 continued on next page*

*Figure 4 continued*

fluorescence during stimulation (seconds 6–14) as depicted in (**A**). Each dot indicates a single animal. Unpaired t-test. **p<0.01. (A+B) Number of animals imaged in N=5 biological replicates: wild type = 40, *flwr-1* = 41. Outliers were removed from both datasets as detected by the iterative Grubb's method (GraphPad Prism). (**C**) Mean (± SEM) body length of animals expressing ChR2(H134R) in GABAergic neurons (*unc-47* promoter) in the *unc-29(e1072)* mutant background, normalized to the average before stimulation. All animals were supplemented with ATR. A 20 s light pulse (470 nm, 100 µW/mm$^2$) was applied after 5 s as indicated by the blue shade. (**D**) Mean (± SEM) relative body length during stimulation (seconds 6–24) as depicted in (**C**). Each dot indicates a single animal. Unpaired t-test. *p<0.05. Number of animals measured in N = 4 biological replicates: wild type = 51, *flwr-1* = 49. (**E**) Mean (± SEM) normalized fluorescence in synaptic puncta of the DNC of animals expressing GCaMP6f and ChrimsonSA in cholinergic motor neurons (*unc-17b* promoter). All animals were supplemented with ATR. A 10 s light pulse (590 nm, 40 µW/mm$^2$) was applied after 5 s as indicated by the red shade. (**F**) Median (with interquartile range [IQR]) normalized fluorescence during stimulation (seconds 6–14) as depicted in (**E**). Each dot indicates a single animal. Kruskal-Wallis test. Only statistically significant differences are depicted. *p<0.05, **p<0.01. (E+F) Number of animals imaged in N = 5 biological replicates: wild type = 40, *flwr-1* = 38, cholinergic rescue = 39, muscle rescue = 36. No outliers were detected by the iterative Grubb's method. (**G**) Schematic representation of motor neuron innervation of body wall muscles (BWMs). Arrows indicate the putatively increased (green) or decreased (red) neurotransmission/excitation of the involved cell types in *flwr-1* mutants compared to wild type.

The online version of this article includes the following source data and figure supplement(s) for figure 4:

**Source data 1.** Raw data for *Figure 4A-F*.

**Figure supplement 1.** Basal SNG-1::pHluorin fluorescence and cell surface fraction are unchanged in *flwr-1* mutants.

**Figure supplement 1—source data 1.** Raw data for *Figure 4—figure supplement 1A and C*.

endocytose fluid from the body cavity, and loss of endocytosis-associated factors affects uptake of proteins secreted from other tissues (*Fares and Grant, 2002*). GFP fused to a secretory signal sequence is discharged from BWMs, and its endocytic uptake by CCs can be quantified by fluorescence microscopy (*Bednarek et al., 2007*; *Fares and Greenwald, 2001*). According to single-cell RNAseq data, FLWR-1 is expressed in CCs (*Supplementary file 3*; *Taylor et al., 2021*). Indeed, mutation of *flwr-1* led to strongly reduced GFP fluorescence levels within CCs, indicating a reduced uptake by endocytosis (*Figure 5A and B*). This defect could be cell-autonomously rescued by expressing FLWR-1 in CCs from the *unc-122* promoter. Furthermore, to assess recycling of SVs, we used the pOpsicle (pH-sensitive optogenetic reporter of synaptic vesicle recycling) assay we recently established (*Seidenthal et al., 2023*) to estimate the amount of SV fusion and the rate of recycling of SV components in cholinergic neurons. This assay combines a pHluorin-based probe fused to an SV-associated protein (SNG-1) and optogenetic stimulation of neurotransmitter release; this way, pHluorin fluorescence is unquenched during stimulated exocytosis and quenched during SV endocytosis and recycling (*Sankaranarayanan et al., 2000*). Since in this assay, signals originating from SVs and from the PM contribute to the overall signal, and since FLWR-1 has an effect on endocytosis, we needed to verify that the relative localization of the SNG-1::pHluorin sensor itself was not affected by the *flwr-1* mutation. Basal fluorescence, before stimulation, was unaltered in *flwr-1* mutants, showing that there is no FLWR-1-dependent alteration of SNG-1::pHluorin in cellular membranes (*Figure 4—figure supplement 1A*). In addition, to estimate the amount of the sensor present in the PM vs. in vesicles, we used primary culture of *C. elegans* cells. Cholinergic neurons expressing SNG-1::pHluorin were imaged and exposed to different pH by adding a buffer of pH 5.6 or by adding ammonium chloride buffer of physiological/neutral pH, which can penetrate the cell and thus shows the maximum achievable signal, i.e., all unquenched pHluorin signal present in the cell. This showed that the amount of SNG-1::pHluorin present in SVs was also not affected by the *flwr-1* mutation (*Figure 4—figure supplement 1B and C*). *In vivo*, loss of FLWR-1 led to significantly increased fluorescence signals during stimulation, which indicates more SV fusion compared to wild type (*Figure 5C and E*). This is in accordance with the increased responses of cholinergic neurons to depolarization we observed earlier (*Figure 4E and F*). Moreover, the rate of fluorescence decay after stimulation was significantly reduced in *flwr-1* mutants, suggesting slower recycling of SVs or reduced acidification of endosomal structures or SVs after recycling (*Figure 5D and F*).

Slowed replenishment of SV pools caused by defective recycling is known to cause synaptic depression, as assessed by electrophysiological measurements of postsynaptic currents (*Kittelmann et al.,*

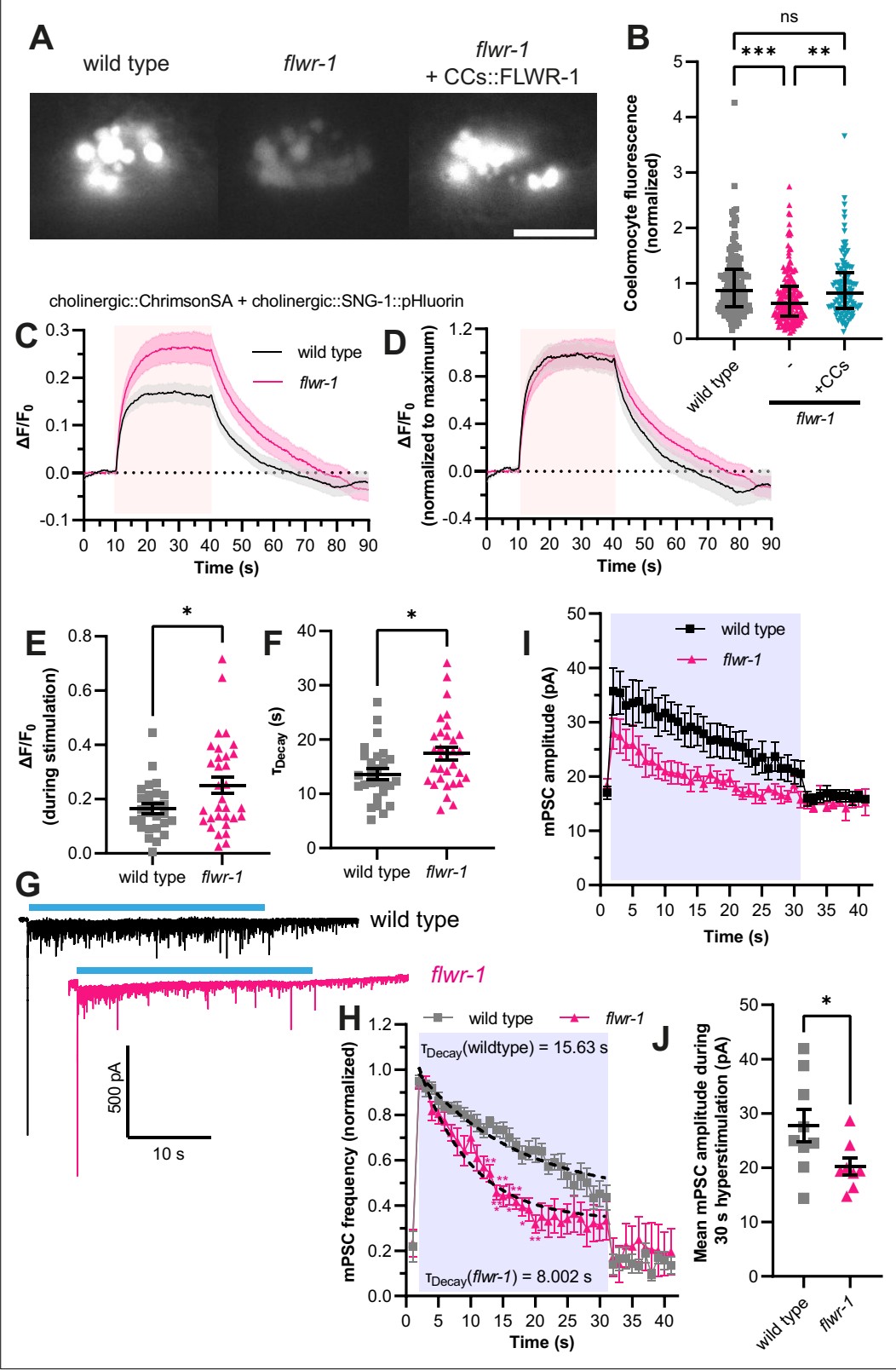

**Figure 5.** FLWR-1 facilitates endocytosis in non-neuronal and neuronal cells. (**A**) Exemplary images of the coelomocytes (CCs) in wild type, *flwr-1(ok3128)* and in *flwr-1* mutants expressing FLWR-1 in CCs (*unc-122* promoter). GFP containing a secretion signal sequence (ssGFP) was expressed in body wall muscles (BWMs) (*myo-3* promoter). Scale bar, 10 μm. (**B**) Median (with interquartile range [IQR]) normalized fluorescence of CCs. Each

*Figure 5 continued on next page*

*Figure 5 continued*

dot indicates a single CC. Kruskal-Wallis test. **p<0.01, ***p<0.001. Number of CCs imaged in N = 3 biological replicates: wild type = 158, *flwr-1* = 185, rescue = 113. (**C**) Mean (± SEM) normalized dorsal nerve cord (DNC) fluorescence of animals expressing SNG-1::pHluorin and ChrimsonSA in cholinergic neurons (*unc-17* promoter). All animals were supplemented with all-*trans* retinal (ATR). A 30 s light pulse (590 nm, 40 µW/mm$^2$) was applied after 10 s as indicated by the red shade. (**D**) Mean (± SEM) pHluorin fluorescence as depicted in (**C**) but additionally normalized to the maximum value of each dataset. (**E**) Mean (± SEM) normalized fluorescence during stimulation (seconds 15–35) as depicted in (**C**). Each dot indicates a single animal. Unpaired t-test. (**F**) Mean (± SEM) calculated exponential decay constants of fluorescence decline after stimulation. Each dot indicates a single animal. Unpaired t-test. (**C–F**) Number of animals imaged in N=5 biological replicates: wild type = 27, *flwr-1* = 32. *p<0.05. (**G**) Representative voltage-clamp recordings of currents in BWMs. Animals expressing ChR2(H134R) in cholinergic motor neurons (*unc-17* promoter, transgene *zxIs6*) were treated with ATR. A 30 s light stimulus (470 nm, 8 mW/mm$^2$) was applied as indicated by blue bars. (**H**) Normalized miniature postsynaptic current (mPSC) frequency in BWMs. All animals were treated with ATR. A 30 s light pulse (470 nm, 8 mW/mm$^2$) was applied as indicated by the blue shade. Dashed lines indicate one-phase exponential regression analysis fitted to the mean mPSC frequencies during stimulation. Calculated time constants of decay are shown. Two-way ANOVA with Šidák's correction. All significant differences to wild type are depicted. (**I**) mPSC amplitude in BWMs of animals measured in (G+H). (**J**) Mean (± SEM) mPSC amplitude during light stimulation as indicated in (**I**). Unpaired t-test. *p<0.05. (**G–J**) Number of animals: wild type = 9, *flwr-1* = 8.

The online version of this article includes the following source data and figure supplement(s) for figure 5:

**Source data 1.** Raw data for *Figure 5B-J*.

**Figure supplement 1.** *flwr-1* mutants show defective cholinergic neurotransmission only during continuous stimulation.

**Figure supplement 1—source data 1.** Raw data for *Figure 5—figure supplement 1A-D, G, H*.

---

*2013b*; *Krick et al., 2021*; *Wu and Betz, 1998*). This is also the case in *Drosophila* for animals lacking Flower (*Yao et al., 2009*). To address this in *C. elegans*, we measured miniature postsynaptic currents (mPSCs; minis) in BWMs (*Figure 5G*), which reflect the postsynaptic effects of presynaptic neurotransmitter release (*Liewald et al., 2008*; *Weissenberger et al., 2011*). Animals lacking FLWR-1 showed no significant difference in basal mPSC frequency or amplitude (*Figure 5H and I*; *Figure 5—figure supplement 1A and B*), which is in accordance with FLWR-1 being dispensable in basal swimming locomotion (*Figure 1C and D*). Similarly, pulsed optogenetic stimulation of cholinergic neurons at different frequencies did not reveal a difference in the measured currents between wild type and mutants (*Figure 5—figure supplement 1C and D*), suggesting that FLWR-1 might be needed only during continuous stimulation. Indeed, *flwr-1* mutants showed an accelerated rundown of the mPSC frequency in BWMs during constant illumination (*Figure 5G and H*). This is in accordance with the role of FLWR-1 in SV recycling. Surprisingly, mPSC amplitudes in *flwr-1* mutants are reduced during 30 s hyperstimulation (*Figure 5I and J*). This suggests that either the amount of neurotransmitter released from a single SV or the number of multivesicular fusion events is decreased (*Liu et al., 2005*). The absolute mPSC frequency, which represents the number of SVs fusing per time, was also smaller during continuous stimulation in *flwr-1* animals (*Figure 5—figure supplement 1E–G*). These results contrast the increased Ca$^{2+}$ responses of cholinergic neurons we observed (*Figure 4E and F*). Since cholinergic neurons also stimulate GABAergic neurons, and since GABAergic minis are also evaluated here, the dissection of animals, required for electrophysiological recordings, could have damaged neuronal commissures, causing an interruption of physiological signal transmission, which is otherwise observed in intact animals. We further analyzed whether *flwr-1* mutants can recover from strong optogenetic 30 s stimulation by applying a short stimulus after a recovery period (interstimulus interval [ISI]) of 15 s and found no significant difference to wild type (*Figure 5—figure supplement 1E and H*). This suggests that a 15 s ISI is sufficient for *flwr-1* mutants to recover to the same extent as wild type synapses. In sum, our results support a facilitatory role of FLWR-1 in SV recycling, specifically during continuous stimulation.

## Loss of FLWR-1 leads to depleted SV pools and accumulation of endocytic structures post-stimulation

Mutants in which endocytosis is affected commonly have fewer SVs because of defective recovery of SV components; this was also found for presynaptic boutons in *Drosophila* mutants lacking Flower (*Schuske et al., 2003*; *Yao et al., 2009*). Such a defect is even more pronounced when samples are conserved immediately following optogenetic stimulation by high-pressure freezing (HPF) (*Kittelmann*

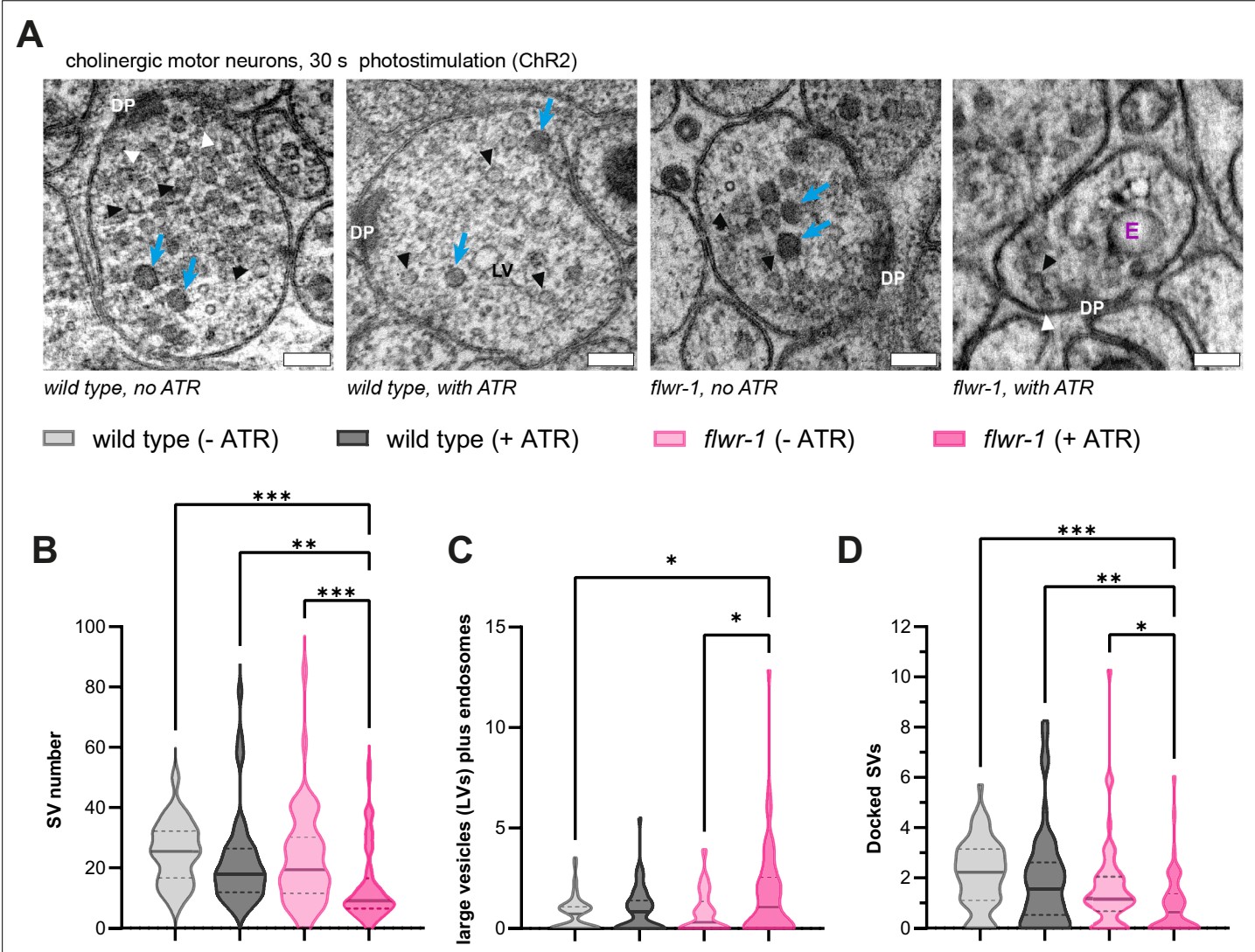

**Figure 6.** Ultrastructural analysis reveals defective recycling of synaptic vesicles (SVs) after stimulation in *flwr-1* mutants. (**A**) Representative transmission electron microscopy (TEM) micrographs of cholinergic en-passant synapses in wild type and *flwr-1(ok3128)* animals expressing ChR2(H134R) in cholinergic neurons (*unc-17* promoter). Animals were optionally treated with all-*trans* retinal (ATR) as indicated. Dense projections (DP), endosomes (abbreviated as E), dense core vesicles (blue arrows), SVs (black arrowheads), docked vesicles (white arrowheads), and large vesicles (LVs) are indicated. Scale bars, 100 nm. (**B**) Violin plot depicting the number of SVs counted per synaptic profile. (**C**) Violin plot depicting the number of large endocytic vesicles and 'endosomes' per synapse. (**D**) Violin plot depicting the number of docked vesicles observed per synaptic profile. (**B–D**) Bold line represents the median, and the dashed lines the interquartile range (IQR). Kruskal-Wallis test. Only statistically significant differences are depicted. *p<0.05, **p<0.01, ***p<0.001. Number of synaptic profiles imaged: wild type (-ATR) = 56, wild type (-ATR) = 51, *flwr-1* (-ATR) = 55, *flwr-1* (+ATR) = 59.

The online version of this article includes the following source data and figure supplement(s) for figure 6:

**Source data 1.** Raw data for *Figure 6B-D*.

**Figure supplement 1.** *flwr-1* mutants have a normal number of dense core vesicles before and after stimulation, but increased size endosomes and large vesicles (LVs).

**Figure supplement 1—source data 1.** Raw data for *Figure 6—figure supplement 1*.

et al., 2013b; Weimer, 2006; Yu et al., 2018). In nematodes, this optogenetic stimulation can be controlled by comparing animals supplemented with ATR, the ChR2 chromophore (Nagel et al., 2005), to animals without ATR. Ultrastructural analysis using transmission electron microscopy (TEM) indeed revealed fewer SVs in stimulated flwr-1 mutant synapses compared to wild type (Figure 6A and B). This indicates that flwr-1 mutants, unlike wild type, are unable to refill SV pools sufficiently fast. At the same time, Drosophila Flower mutants were shown to be defective in the formation of bulk endosomal structures after strong stimulation (Yao et al., 2017). In contrast to this, we observed an increased, rather than decreased, number of endocytic structures (large vesicles [LVs]/'100 nm vesicles') in stimulated flwr-1 mutant synapses (Figure 6C), that were of larger size (Figure 6—figure supplement 1B). This may either be caused by increased SV fusion, which triggers bulk endosomal formation (Clayton et al., 2008; Wu et al., 2014), or defective breakdown of these endocytic structures (Gan and Watanabe, 2018; Watanabe et al., 2013; Yu et al., 2018). Previously, we observed the formation of very large endocytic structures in mutants that affect their resolution into SVs, like endophilin or synaptojanin (Kittelmann et al., 2013b), while a dynamin mutant showed unresolved endocytic structures at, and in continuity with, the PM. Our pHluorin imaging data (Figure 5C–F) would support the notion that both increased SV fusion as well as defective recovery and subsequent acidification of SVs may cause the higher number of endocytic structures in flwr-1 mutants. However, we note that our stimulation regime likely resembles a more physiological activation of neurotransmission compared to the intense stimuli previously used (Yao et al., 2017), as wild type synapses only rarely contained endocytic structures (Kittelmann et al., 2013b). Moreover, we observed fewer docked vesicles in flwr-1 animals which have been treated with ATR, compared to those without (Figure 6D). This might be caused by increased SV fusion and is in line with the increased $Ca^{2+}$ levels during stimulation we observed before. The number of neuropeptide-containing dense core vesicles (DCVs) was unchanged in animals lacking FLWR-1 (Figure 6—figure supplement 1A).

## The increased $Ca^{2+}$ levels of *flwr-1* neurons may be caused by deregulation of MCA-3

While a facilitating role of Flower in endocytosis appears to be conserved in *C. elegans*, in contrast to previous findings from *Drosophila* (Yao et al., 2009), we found no evidence that FLWR-1 conducts $Ca^{2+}$ upon insertion into the PM. On the contrary, presynaptic $Ca^{2+}$ levels were increased during photostimulation of neurons in animals lacking FLWR-1. We thus wondered whether clearance of $Ca^{2+}$, which has entered the synapse via VGCCs, might be defective in *flwr-1* mutants. In neurons and muscles, the PM $Ca^{2+}$ ATPase (PMCA) is involved in extruding $Ca^{2+}$ from the cell (Boczek et al., 2019; Krebs, 2022; Krick et al., 2021). The *C. elegans* homologue MCA-3 (also known as CUP-7) is expressed in neurons, muscle cells, and CCs (Figure 7—figure supplement 1A, Supplementary file 3; Bednarek et al., 2007; Taylor et al., 2021). Interestingly, reducing the function of MCA-3 by mutation was shown to cause a similar defect in endocytosis of secreted GFP in CCs as the loss of FLWR-1 does (Figure 5A and B; Bednarek et al., 2007). We therefore wondered whether MCA-3 might be negatively affected in *flwr-1* mutants. To assess this, we used a mutant *mca-3(ok2048)* lacking part of the C-terminal, regulatory calmodulin-binding domain (C. elegans Deletion Mutant Consortium, 2012; Kraev et al., 1999; Mantilla et al., 2023: Figure 7A). Since *mca-3* loss-of-function mutants are lethal, it is likely that this represents a reduction-of-function mutation (Bednarek et al., 2007). As expected, *mca-3* mutants displayed increased $Ca^{2+}$ influx (or net $Ca^{2+}$ levels, representing the summed VGCC-mediated entry and MCA-3-mediated efflux) upon optogenetic stimulation of cholinergic motor neurons, similar to the effect of the *flwr-1* mutation (Figure 7B and C). Elevated $Ca^{2+}$ levels were not further enhanced in a *flwr-1; mca-3* double mutant. Our data suggest that the two genes are acting in a common pathway. A partially different picture was observed in aldicarb assays, as the reduction of MCA-3 function conveyed aldicarb resistance that was, however, less pronounced than for *flwr-1* mutants (Figure 7D). Nevertheless, since the double mutant showed no exacerbated phenotype compared to the *flwr-1* mutant, both proteins appear to function in the same pathway. If the MCA-3 function was augmented by FLWR-1, then the loss of FLWR-1 may be overcome by higher MCA-3 expression. Since the main focus of FLWR-1 was the GABAergic NMJ (Figure 3D and E), we analyzed aldicarb resistance in *flwr-1* mutants in which we overexpressed MCA-3 in GABAergic neurons (Figure 7—figure supplement 1B). Indeed, MCA-3 overexpression in GABAergic neurons efficiently rescued the *flwr-1* resistance to wild type levels.

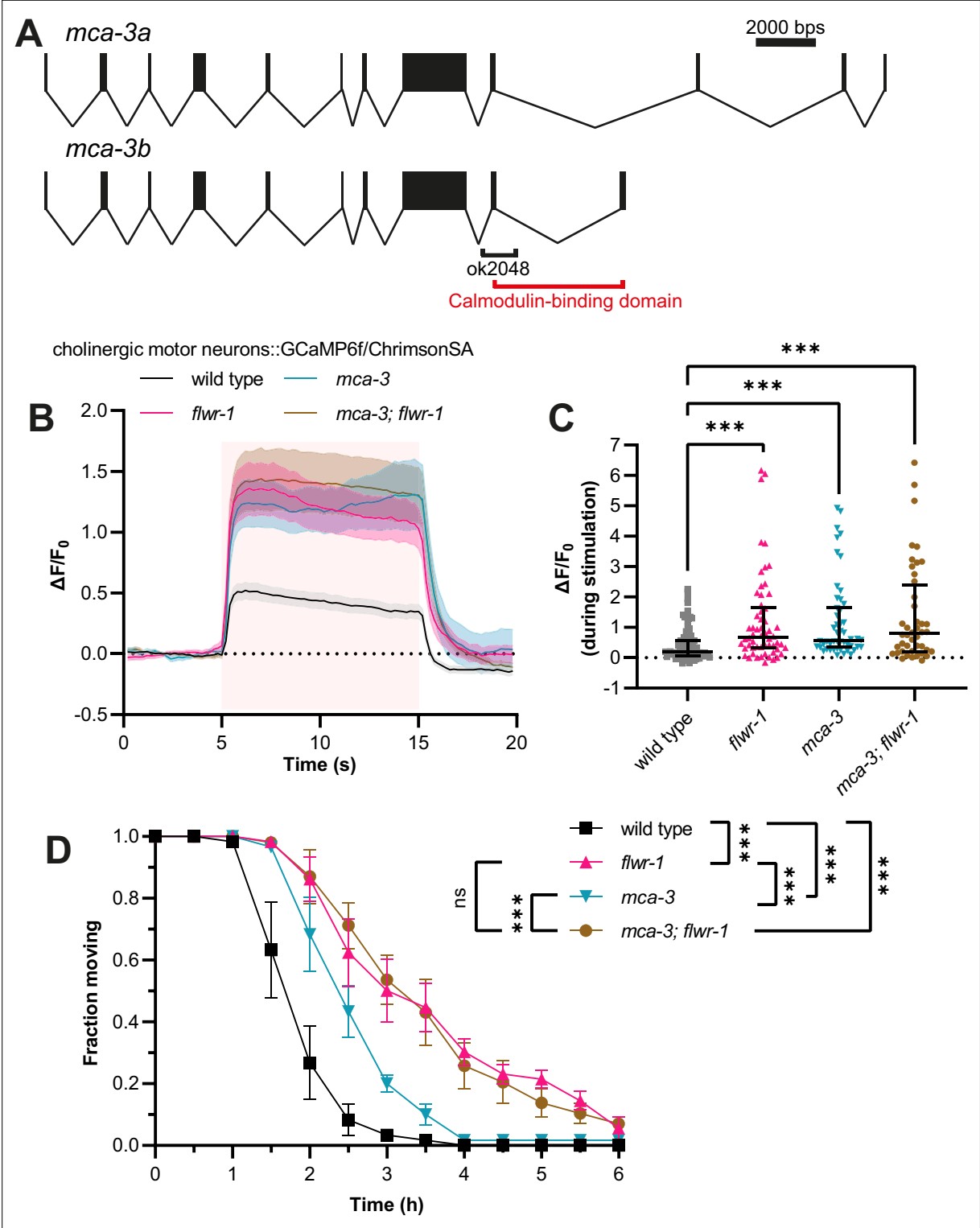

**Figure 7.** Increased Ca²⁺ levels in *flwr-1* mutants may be caused by negative regulation of MCA-3. (**A**) Schematic representation of the *mca-3* gene locus including exon/intron structure of isoforms *mca-3a* and *mca-3b*. Bars represent exons, and connecting lines introns. The size of the *ok2048* deletion as well as the putative calmodulin-binding domain is indicated. (**B**) Mean (± SEM) normalized fluorescence in synaptic puncta of the dorsal nerve cord (DNC) of animals expressing GCaMP6f and ChrimsonSA in cholinergic motor neurons (*unc-17b* promoter). All animals were supplemented with all-*trans* retinal (ATR). A 10 s light pulse (590 nm, 40 µW/mm²) was applied after 5 s as indicated by the red shade. (**C**) Median (with interquartile range [IQR]) normalized fluorescence during stimulation (seconds 6–14) as depicted in (**B**). Each dot indicates a single animal. Kruskal-Wallis test. Only statistically

*Figure 7 continued on next page*

Figure 7 continued

significant differences are depicted. **p<0.01, ***p<0.001. Number of animals imaged in (B+C): wild type = 83, *flwr-1* = 58, *mca-3* = 50, *mca-3; flwr-1* = 44. Outliers were removed from all datasets as detected by iterative Grubb's method (GraphPad Prism). (**D**) Mean (± SEM) fraction of moving animals after exposure to 1.5 mM aldicarb. Two-way ANOVA with Tukey's correction. ns, not significant, ***p<0.001.

The online version of this article includes the following source data and figure supplement(s) for figure 7:

**Source data 1.** Raw data for *Figure 7B-D*.

**Figure supplement 1.** Comparison of *flwr-1* and *mca-3* expression by single-cell RNAseq data.

**Figure supplement 1—source data 1.** Raw data for *Figure 7—figure supplement 1B*.

## FLWR-1 structure prediction does not imply Ca²⁺ conducting ability but PI(4,5)P₂ binding

We observed increased rather than decreased $Ca^{2+}$ levels during stimulation, which contrasts findings from *D. melanogaster* Flower (*Yao et al., 2017*). Previously, an evolutionarily conserved glutamate residue (E78 in the *Drosophila* protein) within the transmembrane domain, which was found to be essential for *Drosophila* Flower function, was suggested to represent a $Ca^{2+}$ selectivity filter because of similarities to $Ca_v1.2$ and TRP channels (*Yao et al., 2009*). This group proposed that Flower, like TRP channels, can form homo-tetramers to conduct $Ca^{2+}$ (*Hoenderop et al., 2003*; *Yao et al., 2009*; *Zhang et al., 2023*). We therefore wondered whether the loss of the conserved glutamate residue impacts FLWR-1 function in *C. elegans* (*Figure 8A*). Expressing a mutant variant of FLWR-1 in which glutamate 74 is exchanged to glutamine (E74Q) significantly decreased the aldicarb resistance of the *flwr-1* mutant (*Figure 8B*), yet did not fully rescue it. This suggests that the respective glutamate is not essential for *C. elegans* FLWR-1 function. To get more insight into the putative structure and oligomerization of FLWR-1, we used AF3 predictions to estimate the approximate location of the conserved glutamate residue within a putative FLWR-1 homo-tetramer (*Abramson et al., 2024*; *Evans et al., 2022*). One of the predicted complexes indeed revealed a pore-like structure (*Figure 8—figure supplement 1A*). However, while conventional $Ca^{2+}$ channels contain a glutamate residue within their pore domain (*Chen et al., 2023*), E74 of FLWR-1 is not predicted to be part of pore-lining residues in the putative tetramer (*Figure 8—figure supplement 1B*).

Flower, like MCA-3, was also suggested to bind to PI(4,5)P₂, an important regulator of endo-/exocytosis and thus SV recycling, and to affect its levels within the PM (*Bednarek et al., 2007*; *Li et al., 2020*; *Lopreiato et al., 2014*). Binding of transmembrane proteins (including TRP channels) to PI(4,5)P₂ is generally mediated by positively charged amino acid residues (lysine and arginine) close to the intracellular side of the PM, which bind to the negatively charged phosphate residues (*Figure 8—figure supplement 1C*; *Duncan et al., 2020*; *Hansen et al., 2011*; *Ribalet et al., 2005*; *Rohacs, 2024*). The predicted FLWR-1 structure similarly contains basic amino acids, which are likely to be close to the intracellular PM leaflet (*Figure 8—figure supplement 1D*). We used in *silico* docking prediction to estimate how PI(4,5)P₂ may bind to FLWR-1. In the resulting structure, both phosphate residues are close to lysine 31 and arginine 27 (*Figure 8—figure supplement 1D*). Positive charges at these positions are evolutionarily conserved, suggesting important functions (*Figure 8A*). Moreover, these amino acids have been suggested to mediate PI(4,5)P₂ binding in *D. melanogaster* Flower (*Li et al., 2020*). Expression of a mutated version of FLWR-1 in which both residues were replaced by alanine only partially rescued aldicarb resistance (*Figure 8B*), implying that FLWR-1 function may be facilitated by these residues and their potential interaction with PI(4,5)P₂. These findings implicate that the conserved E74 is not essential, yet still important for FLWR-1 activity. In addition, an interaction with PI(4,5)P₂ *via* basic amino acids also seems to be an important factor. However, we do note that FLWR-1 structural models as depicted here rely on in *silico* predictions. The actual biological structure of FLWR-1, or its homologues, has yet to be determined using X-ray crystallography or cryo-electron microscopy (cryo-EM).

The pleckstrin homology (PH) domain of phospholipase Cδ (PLCδ) fused to GFP can be used to estimate PI(4,5)P₂ levels within cells (*Botelho et al., 2000*; *Chen et al., 2014*; *Stauffer et al., 1998*). This was used to show that the reduction of MCA-3 function induced decreased PH-PLCδ-GFP fluorescence levels in CCs (*Bednarek et al., 2007*). We thus used this assay to assess the role of FLWR-1 in endosomal PI(4,5)P₂ levels in CCs. Mutation of *flwr-1* reduced the observed GFP fluorescence, suggesting that FLWR-1 positively affects PI(4,5)P₂ levels (*Figure 8C and D*). We further asked if the

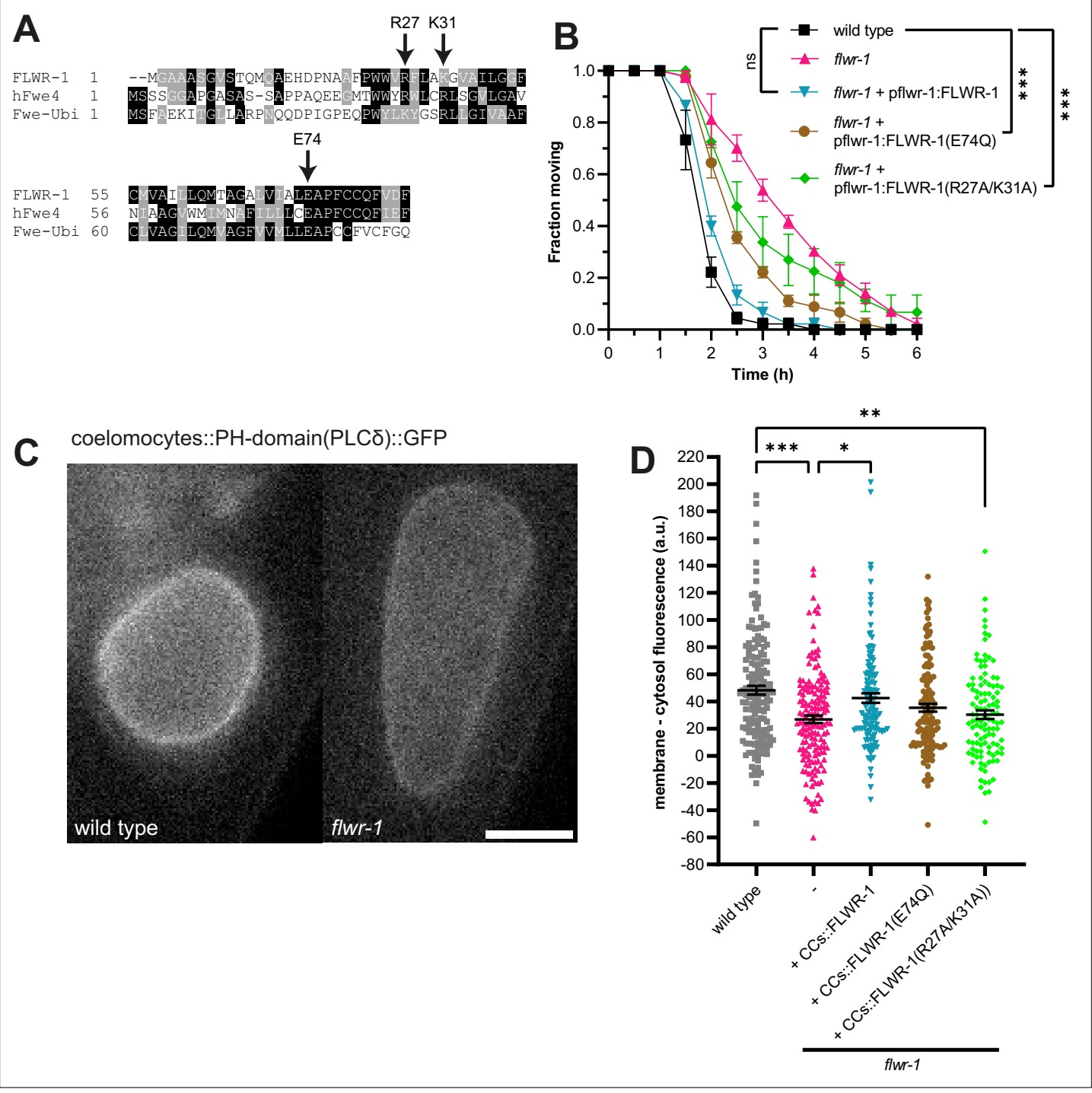

**Figure 8.** Basic amino acid residues on the intracellular surface of FLWR-1 may be involved in PI(4,5)$P_2$ lipid binding. (**A**) Partial alignment of the amino acid sequences of FLWR-1, hFwe4 (*H. sapiens*), and Fwe-Ubi/FweA (*D. melanogaster*). Shading depicts evolutionary conservation of amino acid residues (black – identity; gray – homology). (**B**) Mean (± SEM) fraction of moving animals after exposure to 1.5 mM aldicarb. Two-way ANOVA with Tukey's correction. Selected comparisons are depicted. N=3 biological replicates. ***$p<0.001$. (**C**) Exemplary images of coelomocytes (CCs) in wild type and *flwr-1(ok3128)* animals expressing GFP fused to the PH domain of PLCδ in CCs. Scale bar, 10 μm. (**D**) Mean (± SEM) corrected GFP fluorescence. Membrane fluorescence was estimated by measuring the perimeter of CCs and subtracting the fluorescence in the interior of cells. Each dot indicates a single CC. a.u.=arbitrary units of fluorescence intensity. Kruskal-Wallis test. Only statistically significant differences are depicted. *$p<0.05$, **$p<0.01$, ***$p<0.001$. Number of animals imaged in N=3 biological replicates: wild type = 27, *flwr-1*=25.

The online version of this article includes the following source data and figure supplement(s) for figure 8:

*Figure 8 continued on next page*

*Figure 8 continued*

**Source data 1.** Raw data for *Figure 8B and D*.

**Figure supplement 1.** Structural analysis of important amino acid residues in FLWR-1 and other proteins.

putative PI(4,5)P$_2$ binding site mutations would affect binding of the reporter to the CC membrane. In fact, while expressing FLWR-1 specifically in CCs fully rescued the GFP levels at the CC membrane, expressing the FLWR-1(R27A/K31A) double mutant did not rescue the phenotype, while the E74Q mutant was not significantly different from wild type (*Figure 8C and D*). These findings indicate that FLWR-1 function affects MCA-3 activity through an interaction via PI(4,5)P$_2$ and that FLWR-1 may have an influence on PI(4,5)P$_2$ levels.

## A putative direct interaction of FLWR-1 and MCA-3 may increase MCA-3 function acutely

Next, we asked how FLWR-1 may affect MCA-3 through PI(4,5)P$_2$, and whether this may involve proximity of the two proteins. To test this, we used <u>bi</u>molecular <u>fl</u>uorescence <u>c</u>omplementation (BiFC) *in vivo*. This assay uses two nonfluorescent fragments of mVenus (VN173 and VC155, respectively), which can recombine and complement each other to form a fluorescent protein, as long as they come into close proximity within the cell (*Almedom et al., 2009*; *Chen et al., 2007*; *Hu et al., 2002*). The fragments can be fused to putative interaction partners, and fluorescence complementation (including covalent bond formation in the reconstituted fluorophore) verifies interaction. We thus fused the VC155 fragment to the C-terminus of FLWR-1 and the VN173 fragment to the C-terminus of MCA-3, and co-expressed both proteins (FLWR-1 was expressed from its own promoter, while MCA-3 was expressed in BWM cells; *Figure 9A*). Fluorescence could readily be observed at the PM of muscle cells, but not in other tissues that also express FLWR-1 (*Figure 9B*). As a control, we used two other proteins for probing interactions with FLWR-1: First, the ER-membrane protein NRA-2, which is a homologue of mammalian Nicalin, a component of the translocon-associated Nicalin-TMEM147-NOMO complex that assists folding and assembly of multipass transmembrane proteins in the ER (*McGilvray et al., 2020*; *Smalinskaitė et al., 2022*). We previously showed interactions of Nicalin/NRA-2 and the NOMO homologue NRA-4 by BiFC (*Almedom et al., 2009*). We would expect that FLWR-1, as a multipass TM protein, would also get into close proximity with the NRA-2/Nicalin protein upon biogenesis. Indeed, a clear ER-localized signal was observed when probing FLWR-1::VC155 and NRA-2::VN173 interactions in muscle (*Figure 9B*). Second, to exclude that FLWR-1 would interact with any protein presented in the BiFC assay, we used the UNC-1 stomatin, which is associated with innexin gap junctions in neurons and muscles (*Chen et al., 2007*). Co-expressing FLWR-1::VC155 with UNC-1::VN173 in BWMs, however, did not lead to distinct YFP reconstitution, apart from some general dim fluorescence that was hardly above the level of autofluorescence (*Figure 9B*). These findings indicate that FLWR-1 and MCA-3 may interact with each other in the PM. If this interaction is functional, linking the two proteins via BiFC may not have adverse effects on either protein. We thus tested the transgenic animals in swimming assays (*Figure 9—figure supplement 1*). Neither expression of FLWR-1 with MCA-3 nor FLWR-1 with NRA-2 affected swimming locomotion, showing that there were no dominant negative effects. Note that wild type FLWR-1 and MCA-3 were present in these animals. However, as *mca-3* mutants have a significant reduction in swimming ability (*Figure 9—figure supplement 1*), the fact that FLWR-1/MCA-3 BiFC had no adverse effects indicates that the physical interaction of the two proteins is in line with their native function.

These observations indicate that MCA-3 and FLWR-1 may be interacting directly, and thus their function may jointly be responsible for the Ca$^{2+}$-induced recycling of SVs. Since the increased Ca$^{2+}$ levels in neurons upon stimulation in *flwr-1* mutants may reflect altered MCA-3 abundance at synapses, we tested if FLWR-1 would affect the localization of MCA-3 along neuronal membranes. We expressed MCA-3b::YFP in cholinergic neurons, which was observable along axonal membranes and was enriched in synaptic specializations (*Figure 9C*). In *flwr-1* mutants, this synaptic fluorescence was not overall altered (*Figure 9D*); however, when we compared the YFP signals within the synaptic specializations relative to the inter-synaptic axon shafts, this demonstrated a significantly reduced presence of MCA-3 in synaptic puncta and a relocation to the axonal membranes (*Figure 9E*). This is

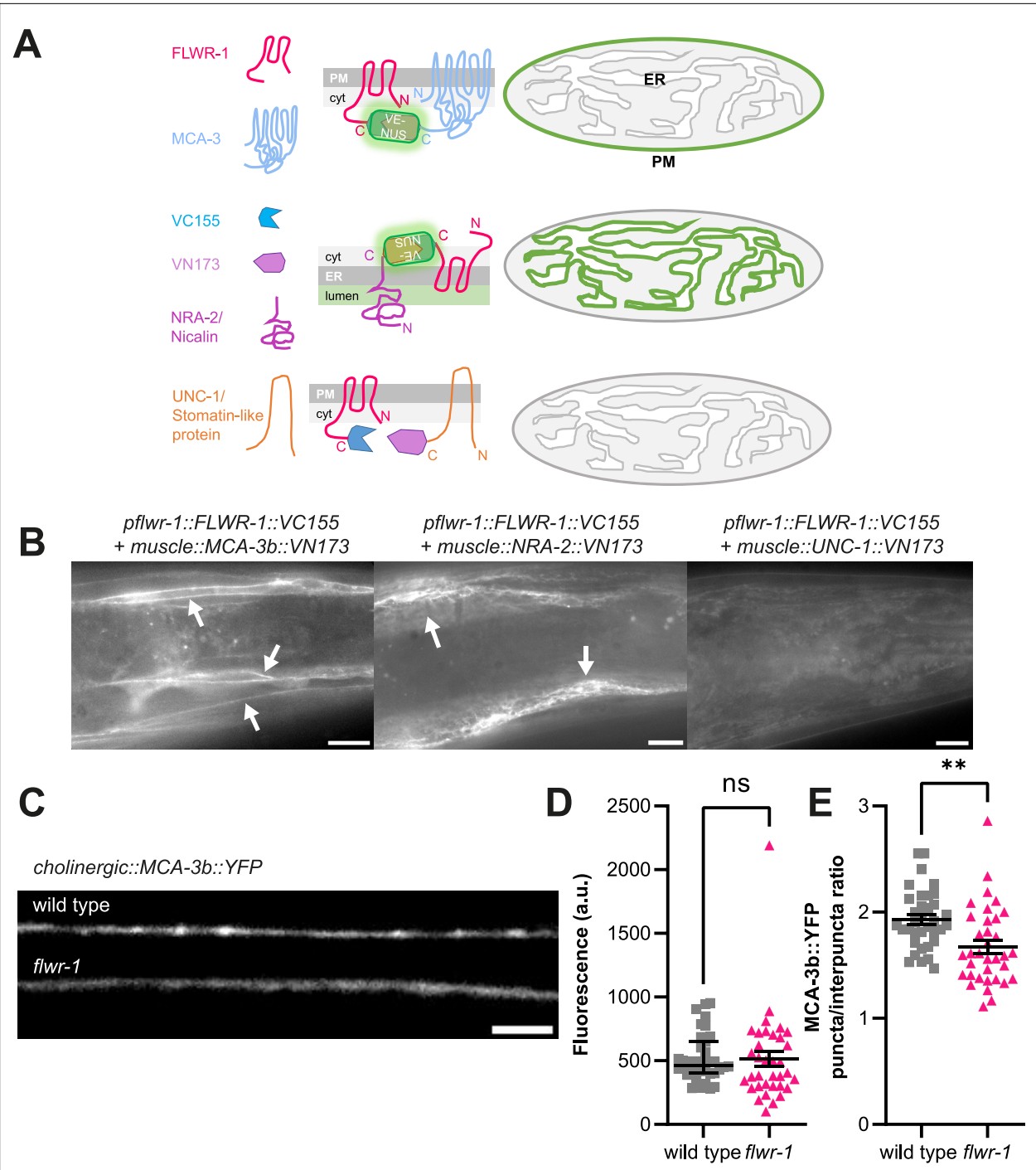

**Figure 9.** FLWR-1 is physically close to MCA-3 in the plasma membrane of body wall muscles (BWMs) and promotes synaptic localization of MCA-3. (**A**) Schematic representation of the <u>bi</u>molecular <u>f</u>luorescence <u>c</u>omplementation (BiFC) assay. (**B**) Representative micrographs of animals expressing FLWR-1 fused to one fragment of mVenus (VN173) and either MCA-3b, NRA-2/Nicalin, or UNC-1 stomatin, interacting with gap junctions, respectively, fused to the other fragment of mVenus (VC155). Scale bar, 10 μm. (**C**) Representative micrographs of the dorsal nerve cord (DNC) of animals expressing MCA-3b::YFP in cholinergic neurons (*unc-17* promoter). Scale bar, 5 μm. (**D, E**) Analysis of MCA-3b::YFP fluorescence (mean ± SEM) along the nerve cord (**D**) and as the ratio of fluorescence in synaptic puncta over fluorescence in the interpuncta regions (**E**). Each dot represents a single animal. Number of animals imaged in N=3 biological replicates: wild type = 35, *flwr-1* = 35. Unpaired t-test. **p<0.01.

The online version of this article includes the following source data and figure supplement(s) for figure 9:

**Source data 1.** Raw data for *Figure 9D and E*.

*Figure 9 continued on next page*

*Figure 9 continued*

**Figure supplement 1.** Expression of FLWR-1 and MCA-3b for <u>bi</u>molecular <u>f</u>luorescence <u>c</u>omplementation (BiFC) does not affect locomotion behavior: mean (±SEM) swimming cycles of animals analyzed by BiFC as depicted in *Figure 8B*.

**Figure supplement 1—source data 1.** Raw data for *Figure 9—figure supplement 1*.

in line with the idea that FLWR-1 function may stabilize/augment MCA-3 expression in the PM and/or its localization in synaptic puncta, to promote efficient synaptic function and SV recycling.

## Discussion

*Drosophila* Flower was shown to affect SV recycling and to alter synaptic $Ca^{2+}$ levels upon stimulation. It was suggested to form a $Ca^{2+}$ channel that gets inserted into the PM upon SV fusion. Here, we showed that also in *C. elegans*, FLWR-1 affects synaptic $Ca^{2+}$ levels following stimulation, that FLWR-1 is localized to SVs in neuronal cells, and that in its absence, endosomal structures with slowed acidification kinetics accumulate, suggesting defective recovery of SVs from recycling endosomes. FLWR-1 may affect SV recycling by stimulating the PMCA MCA-3, possibly by directly interacting with it, and via modulation of $PI(4,5)P_2$ levels, may affect the abundance of MCA-3 in the synaptic membrane. We found that the loss of FLWR-1 conveyed increased $Ca^{2+}$ entry into the motor neuron cytosol, particularly GABAergic neurons, thus leading to more GABA release, an E/I imbalance at the NMJ, and to aldicarb resistance. Since the *C. elegans* NMJ comprises cholinergic and GABAergic neurons, and the aldicarb assay relies on muscle contraction, interpreting results is complicated if a given mutation affects the two motor neuron types (or muscle physiology) differently. We thus included cell-type specific rescue in our analyses. The function of FLWR-1 appears to be ubiquitously involved in membrane trafficking and PM-endosomal recycling, as we found it to affect endocytosis and PM $PI(4,5)P_2$ levels also in CCs. Our findings are summarized in a model in *Figure 10*.

MCA-3 was shown to facilitate endocytosis in CCs (*Bednarek et al., 2007*), indicating that both increased $Ca^{2+}$ levels and defective endocytosis may be influenced by deregulation/lack of stimulation of MCA-3 in *flwr-1* mutants. Whether FLWR-1 directly modulates localization or activates the function of MCA-3, or whether this is caused by a homeostatic change in the absence of FLWR-1, is unclear as yet. Our finding of a putative direct interaction of FLWR-1 with MCA-3 suggests that FLWR-1 could

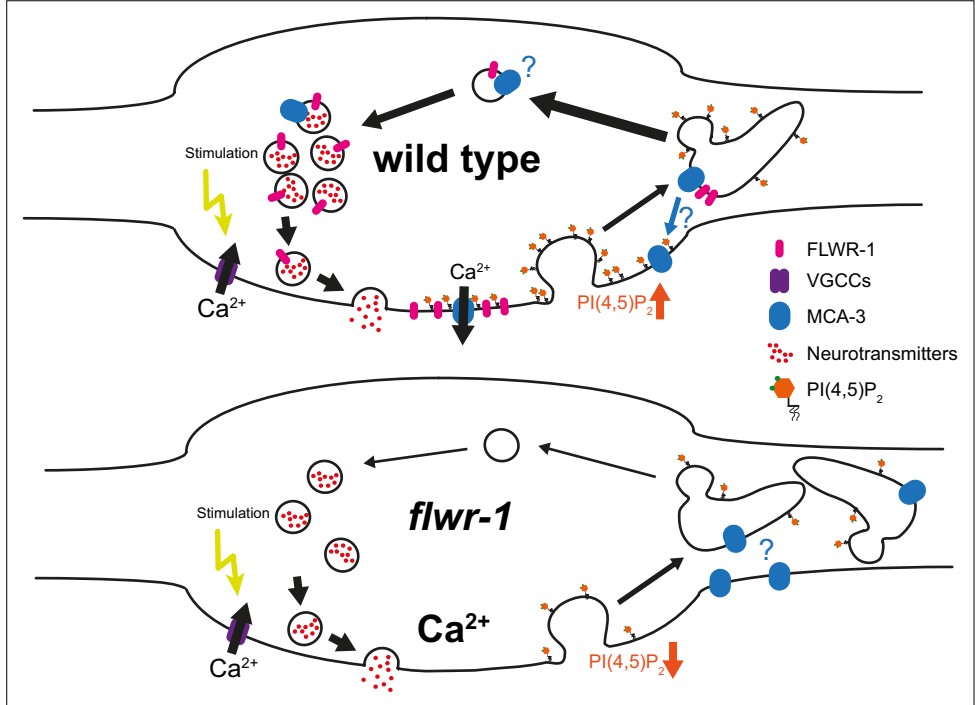

**Figure 10.** Model summarizing findings made in this study. For details, see Discussion.

be inserted into the PM to stimulate the activity of MCA-3 already present in the PM. Alternatively, MCA-3 may undergo dynamic PM insertion and endocytic recycling as part of SVs, along with FLWR-1 (mammalian PMCA was indeed found in purified SVs; *Takamori et al., 2006*). In synapses, this would automatically happen upon SV fusion, such that local $Ca^{2+}$ extrusion at release sites would ensure synapse functionality during intense activity. Such a function of PMCA was shown in *Drosophila*, where PMCA domains delineate areas of SV fusion from areas of SV endocytosis (*Krick et al., 2021*). Thus, independent $Ca^{2+}$ signaling in these two domains, via the P/Q type $Ca^{2+}$ channel $Ca_v2$ (UNC-2 in *C. elegans*) and the L-type $Ca_v1$ (EGL-19 in *C. elegans*), mediating fusion and endocytosis, respectively, is facilitated by the PMCA. MCA-3 could be acutely regulated by FLWR-1, in addition to its regulation by phospholipids (*Lopreiato et al., 2014*). Proof of a direct physical interaction will have to be confirmed by biochemical assays and a biological structure of the putative FLWR-1/MCA-3 complex.

*Drosophila* Flower was shown to increase neuronal $PI(4,5)P_2$ levels, driving ADBE during sustained transmission (*Li et al., 2020*). We could show that an involvement with $PI(4,5)P_2$ is likely conserved in nematodes, requiring the putative $PI(4,5)P_2$ binding site residues R27 and K31 of FLWR-1. Similarly, MCA-3 was important to maintain $PI(4,5)P_2$ in CCs (*Bednarek et al., 2007*). Thus, FLWR-1 and MCA-3 may have a shared role in regulating $PI(4,5)P_2$ levels, or FLWR-1 PM insertion stabilizes local $PI(4,5)P_2$. Recently, another SV protein, synaptotagmin 1 (Syt1), was shown to influence synaptic $PI(4,5)P_2$ levels following fusion, by recruiting PIP kinase I gamma (PIPKIγ) (*Bolz et al., 2023*). Syt1 provides a binding site for PIPKIγ, involving R322 and K326, as part of the sequence **RL**KK**K**. In FLWR-1, R27 and K31 are part of the sequence **R**FLA**K**.

Our results indicate that MCA-3 may be functionally affected/not stimulated in *flwr-1* mutants, explaining increased $Ca^{2+}$ levels during neuronal stimulation (*Brini and Carafoli, 2011*; *Chamberland et al., 2019*; *Ono et al., 2019*). The increased $Ca^{2+}$ level rise we observed was at first surprising, since work in *Drosophila* yielded opposite results during 40 Hz electrical stimulation in neuromuscular boutons of Flower mutants (*Yao et al., 2017*). Upon stimulation with 10 Hz, however, these animals showed an increased GCaMP signal compared to wild type, in line with our observations. Potentially, increased $Ca^{2+}$ in *flwr-1* mutants is only observed with lower, more physiological stimulation. Stimulation by ChrimsonSA, which we used in the $Ca^{2+}$ imaging assays, is not as vigorous; thus, we are likely in a regime comparable to the 10 Hz stimulation in *Drosophila*. Using stronger stimulation via ChR2(H134R), as in the electrophysiological assays, led to a prominent rundown of evoked amplitudes. This may be a consequence of lower $Ca^{2+}$ levels and interference of MCA-3 with UNC-2-dependent $Ca^{2+}$ microdomains required for exocytosis, leading to fewer fusion events. During intense stimulation, larger amounts of PMCA/MCA-3 may be inserted into the PM due to the high number of SVs fusing. Without FLWR-1, less $PI(4,5)P_2$ is generated, and thus MCA-3 may not be effectively recycled from the PM, thus leading to reduced $Ca^{2+}$ levels as observed in *Drosophila*.

Even though we found increased instead of decreased $Ca^{2+}$ level upregulation in *flwr-1* mutants, in line with a stimulatory effect of FLWR-1 on MCA-3, we cannot exclude a possible $Ca^{2+}$ conductivity of FLWR-1. It was discussed that the kinetics of $Ca^{2+}$ levels rising, induced by Flower in *Drosophila*, are too slow and the extent too little as to directly trigger known modes of endocytosis (*Brose and Neher, 2009*; *Leitz and Kavalali, 2016*; *Xue et al., 2012*), e.g., by activating the $Ca^{2+}$ sensor calmodulin (*Wu et al., 2009*). To assess the possibility of FLWR-1 forming an ion channel, we asked AF3 to model FLWR-1 as a tetramer. The putative pore lining residues were mostly hydrophobic or apolar, thus not in agreement with a function as an ion channel. The suggested similarity to the selectivity filter of VGCCs, including the sequence EGW or EAW (in FLWR-1 this is EAP; *Yao et al., 2009*), was not confirmed in the AF3 models: While glutamate faces the extracellular end of the VGCC pore in all four channel modules, $E74_{FLWR-1}$ pointed away from the pore. We thus think that FLWR-1 is unlikely to form an ion channel and suggest that it facilitates SV recycling by stimulating MCA-3 activity.

Activating PMCAs via $PI(4,5)P_2$ (*Berrocal et al., 2017*) would remove $Ca^{2+}$ and prevent sustained phospholipase C activity, thus protecting $PI(4,5)P_2$ at the PM from hydrolysis (*Lopreiato et al., 2014*; *Penniston et al., 2014*) and activating different forms of endocytosis (*Blumrich et al., 2023*; *Bolz et al., 2023*; *Posor et al., 2015*). Such interactions could further trigger a positive feedback loop leading to invagination of the PM (*Yao et al., 2017*). Both PMCAs and Flower were suggested to directly bind $PI(4,5)P_2$ using positively charged amino acid residues close to the membrane (*Filoteo et al., 1992*; *Li et al., 2020*). In line with this, the FLWR-1 structure model exhibits amino acids in suitable positions. This could lead to a close arrangement of both proteins in $PI(4,5)P_2$ lipid

microdomains (*Katan and Cockcroft, 2020*). Flower could also activate PMCAs in intracellular compartments. One PMCA variant was shown to mainly localize to recycling SVs and endosomes, and $Ca^{2+}$ entry into these is an important factor in clearance from the pre-synapse (*Ono et al., 2019*). Overall, this may explain our finding of increased numbers of large endocytic structures in *flwr-1* synapses. Effects of *flwr-1* mutations on $PI(4,5)P_2$ at the PM may negatively affect ADBE, while effects on $PI(4,5)P_2$ in bulk endosomes may slow the breakdown of these structures (*Figure 10*). Our ultrastructural analysis contrasts EM analyses in *Drosophila* which yielded fewer, rather than more, bulk endosomal structures in Flower mutants (*Li et al., 2020*; *Yao et al., 2017*). However, this was following prolonged (10 min) stimulation, possibly resulting in different outcomes compared to our 30 s optogenetic stimulus. Alternatively, the subcellular localization of Flower function may differ between organisms.

Last, Flower could facilitate endocytosis through calcineurin (*Yao et al., 2017*) to stimulate bulk endosome formation. This $Ca^{2+}$- and calmodulin-dependent phosphatase dephosphorylates endocytic proteins to accelerate SV recycling (*Cousin and Robinson, 2001*; *Kumashiro et al., 2005*; *Yamashita, 2012*). How could the increased $Ca^{2+}$ levels in the absence of FLWR-1 and the decreased rate of pHluorin decay fit into this idea? Possibly, increased activation of calcineurin shifts the endocytic mode from ultrafast endocytosis (UFE) (*Watanabe et al., 2013*) to ADBE (*Kittelmann et al., 2013b*; *Yamashita, 2012*). Structures formed by bulk endocytosis are acidified more slowly than small endocytic vesicles generated by UFE, which could explain the delayed decay of pHluorin fluorescence (*Gross and von Gersdorff, 2016*; *Okamoto et al., 2016*). More pronounced neurotransmission, due to increased SV fusion, causes an increase in bulk invaginations (*Clayton et al., 2008*), yet the endocytic structures formed this way are not efficiently disassembled. The *C. elegans* calcineurin TAX-6 is further involved in activity-dependent bulk endocytic processes, yet in a different context, i.e., reorganization of GABAergic synapses during development (*Cuentas-Condori et al., 2023*; *Miller-Fleming et al., 2016*). The remote possibility that FLWR-1 also takes part in this process could imply that developmental aspects of FLWR-1 function may influence E/I balance in adult NMJs.

In sum, our work confirms a conserved role of FLWR-1 in neurotransmission and SV recycling, even though we observed differences to data from *Drosophila* NMJs. We found an unexpected upregulation of neurotransmission in *flwr-1* mutants, which has not yet been observed in other animals and which may result from FLWR-1 acting on PMCAs directly. The observed reduced $Ca^{2+}$ levels in hyperstimulated *Drosophila* synapses may instead result from 'stranded' PMCA in the PM, due to impaired recycling, thus increasing $Ca^{2+}$ extrusion. Further investigations are required to confirm the involvement of FLWR-1 in MCA-3 regulation and the nature of the relationship between these two proteins.

## Materials and methods
### Molecular biology
For plasmids used in this paper and details of their generation, see *Supplementary file 1*. **pZIM902** [punc-17b::GCaMP6fOpt] was a gift from Manuel Zimmer. **p1676** [punc-17(short)::TagRFP::ELKS-1] was kindly provided by the lab of Zhao-Wen Wang. **pJH2523** [punc-25::GCaMP3::UrSL2::wCherry] was a gift from Mei Zhen (Addgene plasmid # 191358; http://n2t.net/addgene:191358; RRID:Addgene_191358) (*Lu et al., 2022*).

### Cultivation of *C. elegans*
Animals were kept at 20°C on nematode growth medium (NGM) plates seeded with OP50-1 bacteria (*Brenner, 1974*). For optogenetic experiments, OP50-1 was supplemented with 200 µM ATR prior to seeding, and animals were kept in darkness. Transgenic animals carrying extrachromosomal arrays were generated by microinjection into the gonads (*Fire, 1986*). L4 staged larvae were picked ~18 hr before experiments and tested on at least 3 separate days with animals picked from different populations. Transgenic animals were selected by fluorescent markers using a Leica MZ16F dissection stereomicroscope. Integration of the *sybIs8965* transgenic array (PHX8965 strain) was performed by SunyBiotech. The RB2305 strain containing the *flwr-1(ok3128)* allele was outcrossed three times to N2 wild type. For an overview of strains, genotypes, and transgenes, see *Supplementary file 2*.

## Counting live progeny

To count living progeny per animal, five L4 larvae were singled onto NGM plates seeded with 50 µl OP50-1. Animals were then picked onto fresh plates after 2 days and on each of the following 2 days and removed on day 5. Living progeny per animal were counted after reaching adulthood and summed up over the three plates on which each parental animal laid eggs, respectively. Experiments were performed blinded to the genotype.

## Pharmacological assays

1.5 mM aldicarb and 0.25 mM levamisole plates were prepared by adding the compounds to liquid NGM prior to plate pouring (*Mahoney et al., 2006*). Aldicarb (Sigma-Aldrich, USA) was kept as a 100 mM stock solution in 70% ethanol. Levamisole (Sigma-Aldrich, USA) was stored as a 200 mM solution dissolved in ddH$_2$O. 12–30 young adult animals were transferred to each plate and tested every 15 (levamisole) or 30 (aldicarb) minutes. Assays were performed blinded to the genotype and control groups (wild type and *flwr-1* mutants), measured in parallel on the same day. Animals that did not respond after being prodded three times with a hair pick were counted as paralyzed. Worms that crawled off the plate were disregarded from analysis.

## Measurement of swimming cycles and crawling speed using the MWT

Swimming cycles were measured as described previously (*Vettkötter et al., 2022*). In short, worms were washed three times with M9 buffer to remove OP50 and transferred onto 3.5 cm NGM plates with 800 µl M9. Animals were visualized using the multiworm tracker (MWT) platform (*Swierczek et al., 2011*) equipped with a Falcon 4M30 camera (DALSA). ChR2 was stimulated with 470 nm light at 1 mW/mm$^2$ intensity for 90 s. 30 s videos were captured, and thrashing was analyzed using the 'wrmTrck' plugin for ImageJ (*Nussbaum-Krammer et al., 2015*). The automatically generated tracks were validated using a custom-written Python script (https://github.com/dvettkoe/SwimmingTracksProcessing; *Vettkötter, 2024a*). Crawling speed was measured with the same setup and as described before (*Vettkötter et al., 2022*), yet animals were transferred to empty, unseeded NGM plates after washing. Prior to the measurement, animals were incubated in darkness for 15 min, which is necessary for a switch from local to global search for food and thus a constant crawling speed (*Calhoun et al., 2014*). The crawling speed was recorded with the 'Multi-Worm Tracker' software and extracted using the 'Choreography' software (*Swierczek et al., 2011*). A custom-written Python script was used to summarize the measured tracks (https://github.com/dvettkoe/MWT_Analysis; *Vettkötter, 2024b*).

For manual counting of swimming cycles (*Figure 9—figure supplement 1*), animals were transferred on an NGM plate in M9 buffer and filmed with a Canon Powershot G11 camera for 30 s. Swimming cycles were counted manually during replay of the video.

## Measurement of body length

Body length was measured as described previously (*Liewald et al., 2008*; *Seidenthal et al., 2022*). In short, single animals were transferred onto unseeded NGM plates and illuminated with light from a 50 W HBO lamp, which was filtered with a 450–490 nm bandpass excitation filter. ChR2 was stimulated with 100 µW/mm$^2$ light intensity. A 665–715 nm filter was used to avoid unwanted activation by brightfield light. Body length was analyzed using the 'WormRuler' software (version 1.3.0) and normalized for the average skeleton length before stimulation (*Seidenthal et al., 2022*). Values deviating more than 20% from the initial body length were discarded as they result from artifacts in the background correction (and are biomechanically impossible). Basal body length was calculated from the skeleton length in pixels, which is generated by the WormRuler software. This was converted to mm and averaged for each animal over a duration of 5 s.

## Light microscopy and fluorescence quantification

Animals were placed upon 7% agarose pads in M9 buffer. Worms were immobilized by either using 20 mM levamisole in M9 or Polybead polystyrene microspheres (Polysciences) for experiments involving GABAergic stimulation. For optogenetic experiments, single animals were placed upon pads to avoid unwanted pre-activation of channelrhodopsins. Imaging was performed using an Axio Observer Z1 microscope (Zeiss, Germany). Proteins were excited using 460 and 590 nm LEDs (Lumen

100, Prior Scientific, UK) coupled via a beamsplitter. Background correction was conducted by placing a region of interest (ROI) within the animal, yet avoiding autofluorescence.

GCaMP and pHluorin imaging and ChrimsonSA stimulation were performed using a 605 nm beamsplitter (AHF Analysentechnik, Germany), which was combined with a double band-pass filter (460–500 and 570–600 nm). A single band-pass emission filter was used (502.5–537.5 nm). pOpsicle assays were performed as described previously using a ×100 objective (*Seidenthal et al., 2023*). We extended the ChrimsonSA stimulation light pulse to 30 s to be consistent with the stimulus length of EM and electrophysiology experiments. Moreover, 2×2 binning was applied. Image sequences with 200 ms light exposure (5 frames per second) were acquired with an sCMOS camera (Kinetix 22, Teledyne Photometrics, USA). Video acquisition was controlled using the µManager v.1.4.22 software (*Edelstein et al., 2014*). The timing of LED activation was managed using an *AutoHotkey* script. DNC fluorescence was quantified with ImageJ by placing a ROI with the *Segmented Line* tool. XY-drift was corrected using the *Template Matching* plugin if necessary. Animals showing excessive z-drift were discarded. A custom-written Python script was used to summarize background subtraction and normalization to the average fluorescence before pulse start (https://github.com/MariusSeidenthal/pHluorin_Imaging_Analysis; *Seidenthal, 2022*).

GCaMP imaging was conducted similarly with the same setup and light intensity for ChrimsonSA stimulation (40 µW/mm$^2$), yet without binning. GCaMP fluorescence was quantified by placing ROIs around four synaptic puncta. Puncta fluorescence was averaged prior to background correction and normalization. Moreover, the bleach correction function of the Python script was used, which corrects fluorescence traces with strong bleaching. Strong bleaching is defined as the fluorescence intensity of the mean of the last second of measurement being less than 85% of the first second. The script performs linear regression analysis for DNC fluorescent traces and background ROI and corrects values prior to background correction. The increase in GCaMP fluorescence during stimulation was highly dependent on basal fluorescence values before stimulation and transient Ca$^{2+}$ signals. This caused some measurements to show very strong increases in fluorescence. To avoid distortion of statistical analyses, we performed outlier detection for the increase during stimulation on all datasets with the GraphPad Prism 9.4.1 Iterative Grubb's method. The alpha value (false discovery rate) was set to 0.01.

For pOpsicle assays, animals were assessed for whether they exhibited a significant increase in fluorescence during stimulation, which was necessary for analysis of fluorescence decay after stimulation. A strong signal was determined as the maximum background corrected fluorescence during the light pulse (moving average of 5 frames) being larger than the average before stimulation plus 3× the standard deviation of the background corrected fluorescence before stimulation. Regression analysis of pHluorin fluorescence decay was performed as described previously by using GraphPad Prism 9.4.1 and fitting a 'Plateau followed one-phase exponential decay' fit beginning with the first time point after stimulation to single measurements (*Seidenthal et al., 2023*). Animals that displayed no increase during stimulation, as well as animals showing no decay or spontaneous signals after stimulation, were discarded from analysis.

Comparison of DNC and VNC fluorescence of GFP::FLWR-1 and mCherry::SNB-1 was conducted by imaging the posterior part of the animal where an abundance of synapses can be found. Images were acquired using the same beamsplitter and excitation filter as used for pOpsicle experiments but equipped with a 500–540 nm/600–665 nm double band-pass emission filter (AHF Analysentechnik, Germany) and 2×2 binning. The Arduino script 'AOTFcontroller' was used to control synchronized two-color illumination (*Aoki et al., 2024*). VNC and DNC fluorescence were analyzed by choosing single frames from acquired z-stacks in which nerve cords were well focused. Fluorescence intensities were quantified by placing a *Segmented Line* ROI. Kymographs were generated with the ImageJ *Multi Kymograph* function.

Confocal laser scanning microscopy was performed on an LSM 780 microscope (Zeiss, Germany) equipped with a Plan-Apochromat ×63 oil objective. The Zeiss Zen (blue edition) *Tile Scan* and *Z-Stack* functions were used to generate overview images of animals expressing GFP::FLWR-1. Images were processed, and maximum z-projection performed using ImageJ. Colocalization analysis was performed by placing a *Segmented Line* ROI onto the DNC and plotting the fluorescence profile along the selected ROI for both color channels. Correlation of fluorescence intensities was then conducted using the GraphPad Prism 9.4.1 Pearson correlation function.

Fluorescence of GFP within CCs was visualized with a ×100 objective. An eGFP filter cube (AHF Analysentechnik, Germany) was used, and fluorescence was excited with the 460 nm LED. Images were acquired using 2×2 binning and 50 ms exposure for ssGFP and 200 ms for GFP fused to the PH domain. Endocytosed ssGFP fluorescence was quantified by drawing a ROI around all CCs found within any worm imaged using the *Freehand selections* tool and performing background correction. CC fluorescence was then normalized to the average fluorescence in wild type animals on the respective measurement day and CC location (anterior/middle/posterior). This was necessary since anterior CCs generally had a higher fluorescence than those in other locations. Membrane localization of PH domain GFP was determined by drawing a *Segmented Line* ROI around the perimeter of the cell and drawing a rough outline of the cell interior using the *Freehand selections* tool. Membrane fluorescence was then corrected by subtracting the measured interior fluorescence for each cell.

In BiFC experiments, split mVenus was imaged with a ×100 objective, the eGFP filter cube, and 500 ms exposure. Fluorescence was excited with 460 nm.

MCA-3b::YFP was imaged with a ×40 objective, the eGFP filter cube, and 200 ms exposure. Fluorescence was excited with 460 nm. Synaptic puncta toward the posterior part of the animal were imaged and quantified by drawing a *Freehand selections* ROI around all visible and well-focused synaptic puncta in an animal before averaging. Similarly, interpuncta fluorescence was quantified by drawing a ROI around several locations in-between the examined puncta.

## Primary neuronal cell culture

Primary neuronal cell cultures were generated from embryos of SNG-1::pHluorin expressing animals as described before (*Seidenthal et al., 2023*), yet seeded into perfusion chambers (0.4 mm channel height, ibidi). pHluorin fluorescence was excited with 460 nm and imaged using the eGFP filter cube and 1000 ms exposure time. The buffers used to quench surface and dequench intracellular pHluorin were adapted from *Dittman and Kaplan, 2006*. Cells were washed three times with the respective buffer prior to imaging by pipetting 1 ml into the reservoir wells of the perfusion chamber and removing the same amount of liquid from the other side. Individual cells were analyzed by placing a *Segmented Line* ROI onto extending neurites and performing background correction. The fraction of SNG-1::pHluorin which resides in SVs was then estimated by dividing the fluorescence originating from vesicular pHluorin by the total fluorescence originating in unquenched pHluorin:

$$Vesicle\,fraction = \frac{F_{NH4Cl} - F_0}{F_{NH4Cl} - F_{pH=5.6}}$$

$F_0 \rightarrow$ the basal neurite fluorescence in control saline buffer
$F_{NH4Cl} \rightarrow$ neurite fluorescence during addition of a buffer containing $NH_4Cl$ which unquenches intravesicular pHluorin
$F_{pH=5.6} \rightarrow$ neurite fluorescence with surface quenched pHluorin by low pH

## Electrophysiology

Electrophysiological recordings of BWMs were performed in dissected adult worms as previously described (*Liewald et al., 2008*). Animals were immobilized with Histoacryl L glue (B. Braun Surgical, Spain), and a lateral incision was made to access NMJs along the anterior VNC. The basement membrane overlying BWMs was enzymatically removed by 0.5 mg/ml collagenase for 10 s (C5138, Sigma-Aldrich, Germany). Integrity of BWMs and nerve cord was visually examined via DIC microscopy.

Recordings from BWMs were acquired in whole-cell patch-clamp mode at 20–22°C using an EPC-10 amplifier equipped with Patchmaster software (HEKA, Germany). The head stage was connected to a standard HEKA pipette holder for fire-polished borosilicate pipettes (1B100F-4, Worcester Polytechnic Institute, USA) of 4–10 MΩ resistance. The extracellular bath solution consisted of 150 mM NaCl, 5 mM KCl, 5 mM $CaCl_2$, 1 mM $MgCl_2$, 10 mM glucose, 5 mM sucrose, and 15 mM HEPES, pH 7.3, with NaOH, ~330 mOsm. The internal/patch pipette solution consisted of 115 mM K-gluconate, 25 mM KCl, 0.1 mM $CaCl_2$, 5 mM $MgCl_2$, 1 mM BAPTA, 10 mM HEPES, 5 mM $Na_2ATP$, 0.5 mM $Na_2GTP$, 0.5 mM cAMP, and 0.5 mM cGMP, pH 7.2, with KOH, ~320 mOsm.

Voltage-clamp experiments were conducted at a holding potential of −60 mV. Light activation was performed using an LED lamp (KSL-70, Rapp OptoElectronic, Hamburg, Germany; 470 nm, 8 mW/mm²) and controlled by the Patchmaster software. Subsequent analysis was performed using

Patchmaster and Origin (OriginLabs). Analysis of mPSCs was conducted with MiniAnalysis (Synaptosoft, Decatur, GA, USA, version 6.0.7), and the rate and amplitude of mPSCs were analyzed in 1 s bins. Exponential decay of mPSCs during stimulation was calculated with GraphPad Prism 9.4.1 by fitting a one-phase exponential decay beginning with the first time point during stimulation.

## Transmission electron microscopy

Prior to HPF, L4 animals were transferred to freshly seeded *Escherichia coli* OP50-1 dishes supplemented with or without 0.1 mM ATR. HPF fixation was performed on young adult animals as described previously (*Kittelmann et al., 2013b*; *Weimer, 2006*). Briefly, 20–40 animals were transferred into a 100 μm deep aluminum planchette (Microscopy Services) filled with *E. coli* (supplemented with or without ATR, respectively), covered with a sapphire disk (0.16 mm) and a spacer ring (0.4 mm; engineering office M Wohlwend) for photostimulation. To prevent pre-activation of ChR2, all preparations were carried out under red light. Animals were continuously illuminated for 30 s with a laser (470 nm, ~20 mW/mm²) followed 5 s later by HPF at –180°C under 2100 bar pressure in an HPM100 (Leica Microsystems). Frozen specimens were transferred under liquid nitrogen into a Reichert AFS machine (Leica Microsystems) for freeze substitution.

Samples were incubated with tannic acid (0.1% in dry acetone) fixative at –90°C for 100 hr. Afterward, a process of washing was performed for substitution with acetone, followed by incubation of the samples in 2% $OsO_4$ (in dry acetone) for 39.5 hr while the temperature was slowly increased up to room temperature. Subsequently, samples were embedded in epoxy resin (Agar Scientific, AGAR 100 Premix kit – hard) by increasing epoxy resin concentrations from 50% to 90% at room temperature and 100% at 60°C for 48 hr.

Electron micrographs of 2–5 individual animals, and from 12 to 22 synapses per treatment, were acquired by cutting cross sections at a thickness of 40 nm, transferring cross sections to formvar- or pioloform-covered copper slot grids. Specimens were counterstained in 2.5% aqueous uranyl acetate for 4 min, followed by washing with distilled water and incubation in Reynolds lead citrate solution for 2 min in a $CO_2$-free chamber with subsequent washing steps in distilled water.

VNC regions were then imaged with a Zeiss 900 TEM, operated at 80 kV, with a Troendle 2 K camera. Images were scored and tagged blind in ImageJ (version 1.53c, National Institute of Health) as described previously and analyzed using SynapsEM (*Vettkötter et al., 2022*; *Watanabe et al., 2020*). Since the number of synaptic organelles varied between synapses of different sizes, their counts were normalized to the average synaptic profile area of 145,253 $nm^2$. LVs are defined by their size larger than 50 nm and empty lumen, as described by *Kittelmann et al., 2013b*. Endosomes are defined as structures larger than 100 nm and located more than 50 nm away from the dense projection, as described by *Watanabe et al., 2014*; however, by an alternative definition of endosome, structures with sizes larger than SVs and DCVs, with an electron-dense lumen, were also included (*Kittelmann et al., 2013b*).

## Protein alignment and structure prediction

Protein alignment was performed with ClustalX (*Larkin et al., 2007*), and shading of conserved residues created with Boxshade (*https://junli.netlify.app/apps/boxshade/*). Transmembrane domains as well as membrane orientation were predicted by entering the predicted amino acid sequence to DeepTMHMM (*Hallgren et al., 2022*). FLWR-1 structure was modeled and visualized with AF3 (*Abramson et al., 2024*). FLWR-1 tetramers have been modeled using AlphaFoldMultimer or AF3 (*Evans et al., 2022*). Protein structures were analyzed and edited using PyMol v2.5.2. Docking of PI(4,5)P₂ to the FLWR-1 AF3 model was conducted in PyMol using the DockingPie plugin running the Autodock/Vina analysis (*Eberhardt et al., 2021*; *Rosignoli and Paiardini, 2022*).

## Statistical analysis and generation of graphs

Graphs were created, and statistical analyses were performed using GraphPad Prism 9.4.1. Data was displayed as mean ± standard error of the mean (SEM) unless noted otherwise. Unpaired t-tests were used to compare two datasets, and one-way ANOVAs or two-way ANOVAs if three or more datasets were assessed, according to the number of tested conditions. Fitting of a mixed-effects model was used instead of two-way ANOVA if the number of data points was not constant between different datasets, for example, if the number of animals differed between time points. Mann-Whitney tests

were performed if two datasets were compared that were not normally distributed, and datasets were depicted as median with interquartile range in statistical analyses. Kruskal-Wallis tests were conducted if more than two not normally distributed datasets were assessed. Iterative Grubb's outlier detection was used when required and as indicated. The alpha value (false discovery rate) was set to 0.01. Representations of the exon-intron structures of genes were created using the 'Exon-Intron Graphic Maker' (*Bhatla, 2012*).

## Acknowledgements

We are indebted to Katharina Kuhlmeier for expert technical assistance and members of the Gottschalk group for critical comments regarding the manuscript. We further thank Martina Rudgalvyte from the Glauser Lab (Université de Fribourg) for helpful comments. We express our gratitude to members of the Zimmer Lab (University of Vienna) for providing advice and plasmids for GCaMP imaging, and to the Wang Lab (University of Connecticut), as well as the Zhen Lab (University of Toronto) for providing plasmids. Some deletion mutations used in this work were provided by the International *C. elegans* Gene Knockout Consortium (*C. elegans* Gene Knockout Facility at the Oklahoma Medical Research Foundation, which is funded by the National Institutes of Health, and the *C. elegans* Reverse Genetics Core Facility at the University of British Columbia, which is funded by the Canadian Institute for Health Research, Genome Canada, Genome BC, the Michael Smith Foundation, and the National Institutes of Health). Finally, we thank the Caenorhabditis Genetics Center (CGC), which is funded by the NIH Office of Research Infrastructure Programs (P40 OD010440), for providing strains. This project was funded by Deutsche Forschungsgemeinschaft, Collaborative Research Centre 1080 project B02 (grants CRC1080/B2 and GO1011/13-2), as well as by core funding from Goethe University.

## Additional information

### Funding

| Funder | Grant reference number | Author |
|---|---|---|
| Deutsche Forschungsgemeinschaft | CRC1080/B2 | Alexander Gottschalk |
| Deutsche Forschungsgemeinschaft | GO1011/13-2 | Alexander Gottschalk |

The funders had no role in study design, data collection and interpretation, or the decision to submit the work for publication.

### Author contributions

Marius Seidenthal, Conceptualization, Resources, Data curation, Formal analysis, Validation, Investigation, Visualization, Methodology, Writing - original draft, Writing – review and editing; Jasmina Redzovic, Data curation, Formal analysis, Investigation; Jana F Liewald, Data curation, Formal analysis, Validation, Investigation, Visualization, Methodology; Dennis Rentsch, Supervision, Validation, Visualization; Stepan Shapiguzov, Investigation; Noah Schuh, Resources, Investigation; Nils Rosenkranz, Investigation, Methodology; Stefan Eimer, Supervision; Alexander Gottschalk, Conceptualization, Funding acquisition, Validation, Visualization, Project administration, Writing – review and editing

### Author ORCIDs

Marius Seidenthal http://orcid.org/0009-0001-0563-7719
Jana F Liewald https://orcid.org/0000-0002-2050-0745
Dennis Rentsch https://orcid.org/0009-0006-9090-9016
Noah Schuh https://orcid.org/0009-0000-1888-7998
Alexander Gottschalk https://orcid.org/0000-0002-1197-6119

Reviewer #1 (Public review): https://doi.org/10.7554/eLife.103870.4.sa1
Reviewer #2 (Public review): https://doi.org/10.7554/eLife.103870.4.sa2
Author response https://doi.org/10.7554/eLife.103870.4.sa3

## Additional files

### Supplementary files

Supplementary file 1. Plasmids used in this study.

Supplementary file 2. Strains used in this study.

Supplementary file 3. CeNGEN expression data of *flwr-1/F20D1.1* and *mca-3* single-cell RNAseq data as described in *Taylor et al., 2021*. The threshold was set to 'All Cells Unfiltered'. Cell types were sorted alphabetically. Where CeNGEN did not produce any results, cells were left blank.

MDAR checklist

### Data availability

All data generated or analysed during this study are included in the manuscript and supporting files; source data files have been provided for the figures.

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
