## [Editor Report · eLife Assessment]

This **important** study uses *C. elegans* to provide new insights into the role of the conserved protein FLWR-1/Flower in synaptic transmission. Employing a variety of techniques, including calcium imaging, ultrastructural analysis, and electrophysiology, the paper provides **convincing** evidence that challenges some previous thinking about FLWR-1 function. This work will be of particular interest to neuroscientists studying synaptic physiology and plasticity.

---

## [Referee Report · Reviewer #1 (Public review)]

The authors investigated the role of the *C. elegans* Flower protein, FLWR-1, in synaptic transmission, vesicle recycling, and neuronal excitability. They confirmed that FLWR-1 localizes to synaptic vesicles and the plasma membrane and facilitates synaptic vesicle recycling at neuromuscular junctions. They observed that hyperstimulation results in endosome accumulation in flwr-1 mutant synapses, suggesting that FLWR-1 facilitates the breakdown of endocytic endosomes. Using tissue-specific rescue experiments, the authors showed that expressing FLWR-1 in GABAergic neurons restored the aldicarb-resistant phenotype of flwr-1 mutants to wild-type levels. By contrast, cholinergic neuron expression did not rescue aldicarb sensitivity at all. They also showed that FLWR-1 removal leads to increased Ca2+ signaling in motor neurons upon photo-stimulation. From these findings, the authors conclude that FLWR-1 helps maintain the balance between excitation and inhibition (E/I) by preferentially regulating GABAergic neuronal excitability in a cell-autonomous manner.

Overall, the work presents solid data and interesting findings, however the proposed cell-autonomous model of GABAergic FLWR-1 function may be overly simplified in my opinion.

Most of my previous comments have been addressed; however, two issues remain.

(1) I appreciate the authors' efforts conducting additional aldicarb sensitivity assays that combine muscle-specific rescue with either cholinergic or GABergic neuron-specific expression of FLWR-1. In the revised manuscript, they conclude, "This did not show any additive effects to the pure neuronal rescues, thus FLWR-1 effects on muscle cell responses to cholinergic agonists must be cell-autonomous." However, I find this interpretation confusing for the reasons outlined below.

Figure 1 - Figure Supplement 3B shows that muscle-specific FLWR-1 expression in flwr-1 mutants significantly restores aldicarb sensitivity. However, when FLWR-1 is co-expressed in both cholinergic neurons and muscle, the worms behave like flwr-1 mutants and no rescue is observed. Similarly, cholinergic FLWR-1 alone fails to restore aldicarb sensitivity (shown in the previous manuscript). These observations indicate a non-cell-autonomous interaction between cholinergic neurons and muscle, rather than a strictly muscle cell-autonomous mechanism. In other words, FLWR-1 expressed in cholinergic neurons appears to negate or block the rescue effect of muscle-expressed FLWR-1. Therefore, FLWR-1 could play a more complex role in coordinating physiology across different tissues. This complexity may affect interpretations of Ca2+ dynamics and/or functional data, particularly in relation to E/I balance, and thus warrants careful discussion or further investigation.

[Editor's note: The authors edited the text of the manuscript to acknowledge potential complexities in the interpretations of these results.]

(2) The revised manuscript includes new GCaMP analyses restricted to synaptic puncta. The authors mention that "we compared Ca2+ signals in synaptic puncta versus axon shafts, and did not find any differences," concluding that "FLWR-1's impact is local, in synaptic boutons." This is puzzling: the similarity of Ca2+ signals in synaptic regions and axon shafts seems to indicate a more global effect on Ca2+ dynamics or may simply reflect limited temporal resolution in distinguishing local from global signals due to rapid Ca2+ diffusion. The authors should clarify how they reached the conclusion that FLWR-1 has a localized impact at synaptic boutons, given that synaptic and axonal signals appear similar. Based on the presented data, the evidence supporting a local effect of FLWR-1 on Ca2+ dynamics appears limited.

[Editor's note: The authors acknowledged that some wording in the previous version was misleading and inaccurate. In the revised version, the authors have withdrawn the conclusion that FLWR-1 function is local in synaptic boutons.]

---

## [Referee Report · Reviewer #2 (Public review)]

Summary:

The Flower protein is expressed in various cell types, including neurons. Previous studies in flies have proposed that Flower plays a role in neuronal endocytosis by functioning as a Ca2+ channel. However, its precise physiological roles and molecular mechanisms in neurons remain largely unclear. This study employs *C. elegans* as a model to explore the function and mechanism of FLWR-1, the *C. elegans* homolog of Flower. This study offers intriguing observations that could potentially challenge or expand our current understanding of the Flower protein. Nevertheless, further clarification or additional experiments are required to substantiate the study's conclusions.

Strengths:

A range of approaches was employed, including the use of a flwr-1 knockout strain, assessment of cholinergic synaptic activity via analyzing aldicarb (a cholinesterase inhibitor) sensitivity, imaging Ca2+ dynamics with GCaMP3, analyzing pHluorin fluorescence, examination of presynaptic ultrastructure by EM, and recording postsynaptic currents at the neuromuscular junction. The findings include notable observations on the effects of flwr-1 knockout, such as increased Ca2+ levels in motor neurons, changes in endosome numbers in motor neurons, altered aldicarb sensitivity, and potential involvement of a Ca2+-ATPase and PIP2 binding in FLWR-1's function.

The authors have adequately addressed most of my previous concerns, however, I recommend minor revisions to further strengthen the study's rigor and interpretation:

Major suggestions

(1) This study relies heavily on aldicarb assays to support its conclusions. While these assays are valuable, their results may not fully align with direct assessment of neurotransmitter release from motor neurons. For instance, prior work has shown that two presynaptic modulators identified through aldicarb sensitivity assays exhibited no corresponding electrophysiological defects at the neuromuscular junction (Liu et al., J Neurosci 27: 10404-10413, 2007). Similarly, at least one study from the Kaplan lab has noted discrepancies between aldicarb assays and electrophysiological analyses. The authors should consider adding a few sentences in the Discussion to acknowledge this limitation and the potential caveats of using aldicarb assays, especially since some of the aldicarb assay results in this study are not easily interpretable.

[Editor's note: The authors added a sentence in the first paragraph of the Discussion to acknowledge these complexities.]

(2) The manuscript states, "Elevated Ca2+ levels were not further enhanced in a flwr-1;mca-3 double mutant." (lines 549-550). However, Figure 7C does not include statistical comparisons between the single and double mutants of flwr-1 and mca-3. Please add the necessary statistical analysis to support this statement.

[Editor's note: In response, the authors noted that these comparisons were indeed carried out. As mentioned in the figure legend, the graph shows only those comparisons that indicated statistical significance.]

(3) The term "Ca2+ influx" should be avoided, as this study does not provide direct evidence (e.g. voltage-clamp recordings of Ca2+ inward currents in motor neurons) for an effect of the flwr-1 mutation of Ca2+ influx. The observed increase in neuronal GCaMP signals in response to optogenetic activation of ChR2 may result from, or be influenced by, Ca2+ mobilization from of intracellular stores. For example, optogenetic stimulation could trigger ryanodine receptor-mediated Ca2+ release from the ER via calcium-induced calcium release (CICR) or depolarization-induced calcium release (DICR). It would be more appropriate to describe the observed increase in Ca2+ signal as "Ca2+ elevation" rather than increased "Ca2+ influx".

[Editor's note: The authors revised their terminology to avoid ambiguities associated with the word "influx".]

---

## [Author Response]

The following is the authors’ response to the previous reviews

**Public Reviews:**

**Reviewer #1 (Public review):**
The authors investigated the role of the *C. elegans* Flower protein, FLWR-1, in synaptic transmission, vesicle recycling, and neuronal excitability. They confirmed that FLWR-1 localizes to synaptic vesicles and the plasma membrane and facilitates synaptic vesicle recycling at neuromuscular junctions. They observed that hyperstimulation results in endosome accumulation in flwr-1 mutant synapses, suggesting that FLWR-1 facilitates the breakdown of endocytic endosomes. Using tissue-specific rescue experiments, the authors showed that expressing FLWR-1 in GABAergic neurons restored the aldicarb-resistant phenotype of flwr-1 mutants to wild-type levels. By contrast, cholinergic neuron expression did not rescue aldicarb sensitivity at all. They also showed that FLWR-1 removal leads to increased Ca^2+^ signaling in motor neurons upon photo-stimulation. From these findings, the authors conclude that FLWR-1 helps maintain the balance between excitation and inhibition (E/I) by preferentially regulating GABAergic neuronal excitability in a cell-autonomous manner.Overall, the work presents solid data and interesting findings, however the proposed cell-autonomous model of GABAergic FLWR-1 function may be overly simplified in my opinion.Most of my previous comments have been addressed; however, two issues remain.(1) I appreciate the authors' efforts conducting additional aldicarb sensitivity assays that combine muscle-specific rescue with either cholinergic or GABergic neuron-specific expression of FLWR-1. In the revised manuscript, they conclude, "This did not show any additive effects to the pure neuronal rescues, thus FLWR-1 effects on muscle cell responses to cholinergic agonists must be cellautonomous." However, I find this interpretation confusing for the reasons outlined below.Figure 1 - Figure Supplement 3B shows that muscle-specific FLWR-1 expression in flwr-1 mutants significantly restores aldicarb sensitivity. However, when FLWR-1 is co-expressed in both cholinergic neurons and muscle, the worms behave like flwr-1 mutants and no rescue is observed. Similarly, cholinergic FLWR-1 alone fails to restore aldicarb sensitivity (shown in the previous manuscript).

This data is still shown in the manuscript, Fig. 3D. We interpreted our finding in the muscle/cholinergic co-rescue experiment as meaning, that FLWR-1 in cholinergic neurons over-compensates, so worms should be resistant, and the rescuing effect of muscle FLWR-1 is therefore cancelled. But it is true, if this were the case, why does the pure cholinergic rescue not show over-compensation? We added a sentence to acknowledge this inconsistency and we added a sentence in the discussion (see also below, comment 1) of reviewer #2.

These observations indicate a non-cell-autonomous interaction between cholinergic neurons and muscle, rather than a strictly muscle cell-autonomous mechanism. In other words, FLWR-1 expressed in cholinergic neurons appears to negate or block the rescue effect of muscle-expressed FLWR-1. Therefore, FLWR-1 could play a more complex role in coordinating physiology across different tissues. This complexity may affect interpretations of Ca^2+^ dynamics and/or functional data, particularly in relation to E/I balance, and thus warrants careful discussion or further investigation.

For the Ca^2+^ dynamics, we think the effects of *flwr-1* are likely very immediate, as the imaging assay relies on a sensor expressed directly in the neurons or muscles under study, and not on indirect phenotypes as muscle contraction and behavior, that depend on an interplay of several cell types influencing each other.

(2) The revised manuscript includes new GCaMP analyses restricted to synaptic puncta. The authors mention that "we compared Ca^2+^ signals in synaptic puncta versus axon shafts, and did not find any differences," concluding that "FLWR-1's impact is local, in synaptic boutons." This is puzzling: the similarity of Ca^2+^ signals in synaptic regions and axon shafts seems to indicate a more global effect on Ca^2+^ dynamics or may simply reflect limited temporal resolution in distinguishing local from global signals due to rapid Ca^2+^ diffusion. The authors should clarify how they reached the conclusion that FLWR-1 has a localized impact at synaptic boutons, given that synaptic and axonal signals appear similar. Based on the presented data, the evidence supporting a local effect of FLWR-1 on Ca^2+^ dynamics appears limited.

We apologize, here we simply overlooked this misleading wording in our rebuttal letter. The data we mentioned, showing no obvious difference in axon vs. bouton, are shown below, including time constants for the onset and the offset of the stimulus (data is peak normalized for better visualization):

One can see that axonal Ca^2+^ signals may rise a bit slower than synaptic Ca^2+^ signals, as expected for Ca^2+^ entering the boutons, and then diffusing out into the axon. The loss of FLWR1 does not affect this. However, the temporal resolution of the used GCaMP6f sensor is ca. 200 ms to reach peak, and the decay time (to t1/2) is ca. 400 ms (PMID: 23868258). Thus, it would be difficult to see effects based on Ca^2+^ diffusion using this assay. For the decay, this is similar for both axon and synapse, while *flwr-1* mutants do not reduce Ca^2+^ as much as wt. In the axon, there is a seemingly slightly slower reduction in *flwr-1* mutants, however, given the kinetics of the sensor, this is likely not a meaningful difference. Therefore, we wrote we did not find differences. The interpretation should not have been that the impact of FLWR-1 is local. It may be true if one could image this at faster time scales, i.e. if there is more FLWR-1 localized in boutons (as indicated by our data showing FLWR-1 enrichment in boutons; Fig. 3), and when considering its possible effect on MCA-3 localization (and assuming that MCA-3 is the active player in Ca^2+^ removal), i.e. FLWR-1 recruiting MCA-3 to boutons (Fig. 9C, D).

**Reviewer #2 (Public review):**
Summary:The Flower protein is expressed in various cell types, including neurons. Previous studies in flies have proposed that Flower plays a role in neuronal endocytosis by functioning as a Ca^2+^ channel. However, its precise physiological roles and molecular mechanisms in neurons remain largely unclear. This study employs *C. elegans* as a model to explore the function and mechanism of FLWR-1, the C. elegans homolog of Flower. This study offers intriguing observations that could potentially challenge or expand our current understanding of the Flower protein. Nevertheless, further clarification or additional experiments are required to substantiate the study's conclusions.Strengths:A range of approaches was employed, including the use of a flwr-1 knockout strain, assessment of cholinergic synaptic activity via analyzing aldicarb (a cholinesterase inhibitor) sensitivity, imaging Ca^2+^ dynamics with GCaMP3, analyzing pHluorin fluorescence, examination of presynaptic ultrastructure by EM, and recording postsynaptic currents at the neuromuscular junction. The findings include notable observations on the effects of flwr-1 knockout, such as increased Ca^2+^ levels in motor neurons, changes in endosome numbers in motor neurons, altered aldicarb sensitivity, and potential involvement of a Ca^2+^-ATPase and PIP2 binding in FLWR-1's function.The authors have adequately addressed most of my previous concerns, however, I recommend minor revisions to further strengthen the study's rigor and interpretation:Major suggestions(1) This study relies heavily on aldicarb assays to support its conclusions. While these assays are valuable, their results may not fully align with direct assessment of neurotransmitter release from motor neurons. For instance, prior work has shown that two presynaptic modulators identified through aldicarb sensitivity assays exhibited no corresponding electrophysiological defects at the neuromuscular junction (Liu et al., J Neurosci 27: 10404-10413, 2007). Similarly, at least one study from the Kaplan lab has noted discrepancies between aldicarb assays and electrophysiological analyses. The authors should consider adding a few sentences in the Discussion to acknowledge this limitation and the potential caveats of using aldicarb assays, especially since some of the aldicarb assay results in this study are not easily interpretable.

Aldicarb assays have been used very successfully in identifying mutants with defects in chemical synaptic transmission, and entire genetic screens have been conducted this way. The reviewer is right, one needs to realize that it is the balance of excitation and inhibition at the NMJ of *C. elegans*, which underlies the effects on the rate of aldicarb-induced paralysis, not just cholinergic transmission. I.e. if a given mutant affects cholinergic and GABAergic transmission differently, things become difficult to interpret, particularly if also muscle physiology is affected. Therefore, we combined mutant analyses with cell-type specific rescue. We acknowledge that results are nonetheless difficult to interpret. We thus added a sentence in the first paragraph of the discussion.

(2) The manuscript states, "Elevated Ca^2+^ levels were not further enhanced in a flwr-1;mca-3 double mutant." (lines 549-550). However, Figure 7C does not include statistical comparisons between the single and double mutants of flwr-1 and mca-3. Please add the necessary statistical analysis to support this statement.

Because we only marked significant differences in that figure, and n.s. was not shown. This was stated in the figure legend.

(3) The term "Ca^2+^ influx" should be avoided, as this study does not provide direct evidence (e.g. voltage-clamp recordings of Ca^2+^ inward currents in motor neurons) for an effect of the flwr-1 mutation of Ca^2+^ influx. The observed increase in neuronal GCaMP signals in response to optogenetic activation of ChR2 may result from, or be influenced by, Ca^2+^ mobilization from of intracellular stores. For example, optogenetic stimulation could trigger ryanodine receptor-mediated Ca^2+^ release from the ER via calcium-induced calcium release (CICR) or depolarization-induced calcium release (DICR). It would be more appropriate to describe the observed increase in Ca^2+^ signal as "Ca^2+^ elevation" rather than increased "Ca^2+^ influx".

Ok, yes, we can do this, we referred by ‘influx’ to cytosolic Ca^2+^, that fluxes into the cytosol, be it from the internal stores or the extracellular. Extracellular influx, more or less, inevitably will trigger further influx from internal stores, to our understanding. We changed this to “elevated Ca^2+^ levels” or “Ca^2+^ level rise” or “Ca^2+^ level increase”.

**Recommendations for the authors:**

**Reviewer #1 (Recommendations for the authors):**
A thorough discussion on the impact of cell-autonomous versus non-cell-autonomous effects is necessary.Revise and clarify the distinction between local and global Ca²⁺ changes.

see above.

**Reviewer #2 (Recommendations for the authors):**
Minor suggestions(1) In "Few-Ubi was shown to facilitate recovery of neurons following intense synaptic activity Yao et al.,....." (lines 283-284), please specify which aspects of neuronal recovery are influenced by the Flower protein.

We added “refilling of SV pools”.

(2) The abbreviation "Few-Ubi" is used for the *Drosophila* Flower protein (e.g., line 283, Figure 1A, and Figure 8A). Please clarify what "Ubi" stands for and verify whether its inclusion in the protein name is appropriate.

This is inconsistent across the literature, sometimes Fwe-Ubi is also referred to as FweA. We now added this term. Ubi refers to ubiquitous (“Therefore, we named this isoform fweubi because it is expressed ubiquitously in imaginal discs“) (Rhiner 2010)

(3) The manuscript uses "pflwr-1" (line 303 and elsewhere) to denote the flwr-1 promoter. This notation could be misleading, as it may be interpreted as a gene name. Please consider using either "flwr-1p" or "Pflwr-1" instead. Additionally, ensure proper italicization of gene names throughout the manuscript.

We changed this throughout. We will change to italicized at proof stage, it would be too timeconsuming to spot these incidents now.

(4) The authors tagged the C-terminus of FLWR-1 by GFP (lines 321). The fusion protein is referred to as "GFP::FLWR-1" throughout the manuscript. Please verify whether "FLWR-1::GFP" would be the more appropriate designation.

Thank you, yes, we changed this in the text, GFP is indeed N-terminal.

(5) In "This did not show any additive effects...." (line 363), please clarify what "This" refers to.

Altered to “The combined rescues did not show any additive effects…”

(6) In "..., supporting our previous finding of increased neurotransmitter release in GABAergic neurons" (lines 412-413), please provide a citation for the referenced previous study.

This refers to our aldicarb data within this paper, just further up in the text. We removed “previous”.

(7) Figure 4C, D examines the effect of flwr-1 mutation on body length in the genetic background of the unc-29 mutation, which selectively disrupts the levamisole-sensitive acetylcholine receptor. Please comment on the rationale for implicating only the levamisole receptor rather than the nicotinic acetylcholine receptor in muscle cells.

This was because we used a behavioral assay. Despite the fact that the homopentameric ACR16/N-AChR mediate about 2/3 of the peak currents in response to acute ACh application to the NMJ (e.g. Almedom et al., EMBO J, 2009), the *acr-16* mutant has virtually no behavioral / locomotion phenotype. Likely, this is because the heteropentameric, UNC-29 containing LAChR, while only contributing 1/3 of the peak current, desensitizes much more slowly and thus *unc-29* mutants show a severe behavioral phenotype (uncoordinated locomotion, etc.). We thus did not expect a major effect when performing the behavoral assay in *acr-16* mutants and thus chose the *unc-29* mutant background.

(8) In "we found no evidence ....insertion into the PM (Yao et al., 2009)", It appears that the cited paper was not authored by any of the current manuscript. Please confirm whether this citation is correctly attributed.

This sentence was arranged in a misleading way, we did not mean that we authored this paper. It was change in the text: “While a facilitating role of Flower in endocytosis appears to be conserved in *C. elegans*, in contrast to previous findings from *Drosophila* (Yao et al., 2009), we found no evidence that FLWR-1 conducts Ca^2+^ upon insertion into the PM.”